# Development of mirror-image monobodies targeting the oncogenic BCR::ABL1 kinase

Nina Schmidt ®[1], Amit Kumar[1], Lukas Korf[2], Adrian Valentin Dinh-Fricke[1], Frank Abendroth[3], Akiko Koide ®[4,5], Uwe Linne[6], Magdalena Rakwalska-Bange[1], Shohei Koide ®[5,7], Lars-Oliver Essen[2], Olalla Vázquez[3,8] ✉ & Oliver Hantschel ®[1] ✉

Mirror-image proteins, composed of D-amino acids, are an attractive therapeutic modality, as they exhibit high metabolic stability and lack immunogenicity. Development of mirror-image binding proteins is achieved through chemical synthesis of D-target proteins, phage display library selection of L-binders and chemical synthesis of (mirror-image) D-binders that consequently bind the physiological L-targets. Monobodies are well-established synthetic (L-) binding proteins and their small size (~90 residues) and lack of endogenous cysteine residues make them particularly accessible to chemical synthesis. Here, we develop monobodies with nanomolar binding affinities against the D-SH2 domain of the leukemic tyrosine kinase BCR::ABL1. Two crystal structures of heterochiral monobody-SH2 complexes reveal targeting of the pY binding pocket by an unconventional binding mode. We then prepare potent D-monobodies by either ligating two chemically synthesized D-peptides or by self-assembly without ligation. Their proper folding and stability are determined and high-affinity binding to the L-target is shown. D-monobodies are protease-resistant, show long-term plasma stability, inhibit BCR::ABL1 kinase activity and bind BCR::ABL1 in cell lysates and permeabilized cells. Hence, we demonstrate that functional D-monobodies can be developed readily. Our work represents an important step towards possible future therapeutic use of D-monobodies when combined with emerging methods to enable cytoplasmic delivery of monobodies.

Synthetic binding proteins with high affinity and selectivity can be engineered from natural immunoglobulin scaffolds (e.g., scFvs, Fabs, nanobodies) or non-immunoglobulin scaffolds (e.g., monobodies, DARPins, affibodies, anticalins)[1–3]. They are broadly used as research tools in structural-, cell- and molecular biology[4]. Some protein binder classes, such as DARPins, monobody and its analuegs, affibodies and anticalins, moved to clinical development stage for the treatment of cancer and other diseases. Their small size (~8–20 kDa), rapid generation by molecular display techniques, high-affinity target binding and facile recombinant production

[1]Institute of Physiological Chemistry, Faculty of Medicine, Philipps University of Marburg, Marburg, Germany. [2]Faculty of Chemistry and Unit for Structural Biology, Philipps University of Marburg, Marburg, Germany. [3]Faculty of Chemistry and Unit for Chemical Biology, Philipps University of Marburg, Marburg, Germany. [4]Department of Medicine, New York University School of Medicine, New York, NY, USA. [5]Laura and Isaac Perlmutter Cancer Center, New York University Langone Health, New York, NY, USA. [6]Faculty of Chemistry and Unit for Mass Spectrometry, Philipps University of Marburg, Marburg, Germany. [7]Department of Biochemistry and Molecular Pharmacology, New York University School of Medicine, New York, NY, USA. [8]Center for Synthetic Microbiology (SYNMIKRO), Philipps University of Marburg, Marburg, Germany. ✉e-mail: olalla.vazquez@staff.uni-marburg.de; oliver.hantschel@uni-marburg.de

enables targeting of a broad range of diverse proteins and promises great therapeutic opportunities[5].

Among the most commonly used non-immunoglobulin binders are monobodies (Mb), which are generated from large combinatorial libraries using a fibronectin III domain (FN3; ~10 kDa) of human fibronectin as molecular scaffold[6,7]. Since their first report[6], pharma and biotech industries developed monobody analogues, termed adnectins, tenascins and centyrins, which target extracellular proteins and receptors, such as VEGFR2, EGFR, PCSK9 and were evaluated in phase I-II clinical trials[3,4,8,9]. We have developed monobodies that target major (intracellular) oncogenes, including kinases (BCR::ABL1, SRC), phosphatases (SHP-2), transcription factors (STAT3), chromatin readers (WDR5) and small GTPases (H-/K-RAS)[10–18]. Monobodies to other targets, such as the SARS-CoV-2 spike protein and optineurin were recently developed by others[19,20].

Upon genetic expression of monobodies in tumor cell lines and primary cells, we and others observed selective inhibition of oncoprotein-dependent signaling[11–13,15,16]. We are developing technologies for intracellular monobody protein delivery and, in parallel, improved pharmacokinetics and biodistribution of monobodies[2,21–23]. For therapeutic translation, a high plasma and intracellular stability is an important prerequisite and of hallmark importance to enable in vivo application and efficacy of monobodies. Therefore, we set out to develop mirror-image D-monobodies.

Mirror-image proteins that are formed from amino acids in D-configuration exhibit several advantages compared to naturally occurring L-proteins: Higher in vivo pharmacological stability, such as higher plasma half-life, is due to protease resistance, as D-peptide bonds cannot be cleaved by natural proteases. In addition, D-proteins display very low immunogenicity, as immune cells cannot proteolytically generate D-peptides, and even if fragments were generated, they would not be able to be displayed on the major histocompatibility complex (MHC)[24–26].

A common approach to develop D-binding proteins is the so-called mirror-image phage display, which circumvents the challenge of directly developing such binding proteins using a D-molecular display technology[27]. First, a relevant target protein is produced in the D-configuration through solid-phase peptide synthesis (SPPS), followed by stepwise native chemical ligation (NCL) of the peptides and refolding to obtain the folded mirror-image target protein[28,29]. This D-protein is then subjected to a selection process with the combinatorial L-protein binder libraries. As the selected L-binders provide a blueprint for D-protein binders to the natural L-target protein, the synthesized D-binder (by SPPS, NCL and refolding) will consequently, by symmetry, bind to the natural L-target (Fig. 1). This workflow was applied to develop D-peptide antagonists of different protein-protein interactions[27,30–33].

While D-peptides can be readily made by SPPS, D-protein binder synthesis requires the same tedious process consisting of SPPS, NCL of peptides and subsequent refolding as production of the D-target protein for screening. Although pioneering work by the Kent and Sidhu groups reported mirror-image GB1 protein variants (56 amino acids) targeting VEGF-A, to our knowledge, their approach has not resulted in a therapeutic application or development of protein binders in D-configuration to other target proteins to date[24,25]. Other well-established engineered binding proteins, including antibody fragments and DARPins have not been reported in the D-form so far. Our previous work demonstrated the successful chemical synthesis of the BCR::ABL1 SH2 domain[34] and a proof-of-concept study showed the synthesis of a D-monobody, but without target binding function, using a 3 peptide ligation strategy[26]. To explore the feasibility of D-monobody development and assess its full potential, we set out to develop a more facile strategy to synthesize D-monobodies and chose a therapeutically important intracellular target, the SH2 domain of BCR::ABL1. The Philadelphia chromosomal translocation produces a

fusion of the breakpoint cluster region (BCR) and the Abelson tyrosine kinase (ABL1) genes, which results in the expression of the BCR::ABL1 protein. In chronic myeloid leukemia (CML) and a large fraction of B-cell acute lymphoblastic leukemias, the constitutive tyrosine kinase activity of BCR::ABL1 is the central driver of leukemogenesis[35]. While different BCR::ABL1 tyrosine kinase inhibitors (TKIs) have strongly improved the overall survival of most CML patients, TKI-resistance and -intolerance result in disease recurrence and prevent cure[35,36]. We previously developed a series of (L-)monobodies directed to the Src homology 2 (SH2) domain of BCR::ABL1, which resulted in inhibition of BCR::ABL1 activity, signaling and leukemogenesis in CML cells through different mechanisms-of-action[10,11,13]. SH2 domains are modular protein-protein interaction domains with 120 members in humans and bind tyrosine-phosphorylated peptide sequences with moderate selectivity and affinities in the low micromolar range[37]. Despite being important drug targets, selective inhibition of SH2 domains by high-affinity peptides, peptidomimetics or small molecules are notoriously difficult to develop, although some good progress was made recently[38–41]. While BCR::ABL1 is an intracellular protein and no established technologies for efficient cytosolic delivery of D-monobodies are available yet, we still decided to follow this ambitious path, which will enable the rapid assessment of delivered D-monobody proteins once fully validated delivery approaches, which we and others are developing, become available in the near future[2]. Furthermore, BCR::ABL1 is an established therapeutic target and our lab has made important contributions to the BCR::ABL1 field[11,18,42–45]. Therefore, many tools and assays for the characterization of BCR::ABL1-targeting monobodies are readily available in our lab and we are able to compare newly generated monobodies with previously published ones[10,13].

Here, we show the development of two D-monobodies targeting the BCR::ABL1 SH2 domain with high affinity and their structural characterization by solving the crystal structures of heterochiral monobody-SH2 complexes. For functional characterization, one of the D-monobodies is refolded to its native mirror-conformation after NCL of two D-peptide segments and subsequent desulfurization. For the other D-monobody clone, NCL is omitted and a functional split-D-monobody is created by refolding a 1:1 mixture of two peptides spanning the entire length of the monobody. Subsequent functional characterization shows high-affinity target binding, protease resistance, high plasma stability, inhibition of BCR::ABL1 kinase activity and binding to BCR::ABL1 in CML cell lysates and permeabilized cells. We also refer to a parallel study by Hayashi et al. that develops a D-monobody targeting the inflammation-related cytokine MCP-1 using a similar workflow[46]. Our two studies utilize different but complementary approaches and demonstrate the facile development of a functional mirror-image D-monobody targeting a therapeutically important protein. This convergence suggests the robustness of the platforms and provides strong evidence for the feasibility and potential impact of mirror-image monobody binders.

## Results
### Generation of L-monobodies targeting the D-Abl SH2 domain
We first produced the D-version of the human BCR::ABL1 SH2 domain (Abl SH2) by native chemical ligation (NCL) of two D-peptide fragments obtained by solid-phase peptide synthesis (SPPS). The cysteine residue required for NCL was subsequently desulfurized to an alanine residue to yield the wildtype sequence (Fig. 1)[34]. As previously described, the protein was monomeric in size exclusion chromatography (SEC) after refolding, showed a similar secondary structure composition and thermal stability as the recombinant Abl SH2 domain, and bound a D-phospho tyrosine (pY) peptide with the same affinity as the L-counterpart to the corresponding L-pY peptide[34].

We selected monobodies from the combinatorial "side-and-loop" phage-display library and subsequent yeast display screening (Supplementary information, section 2)[7]. Monobody pools enriched with

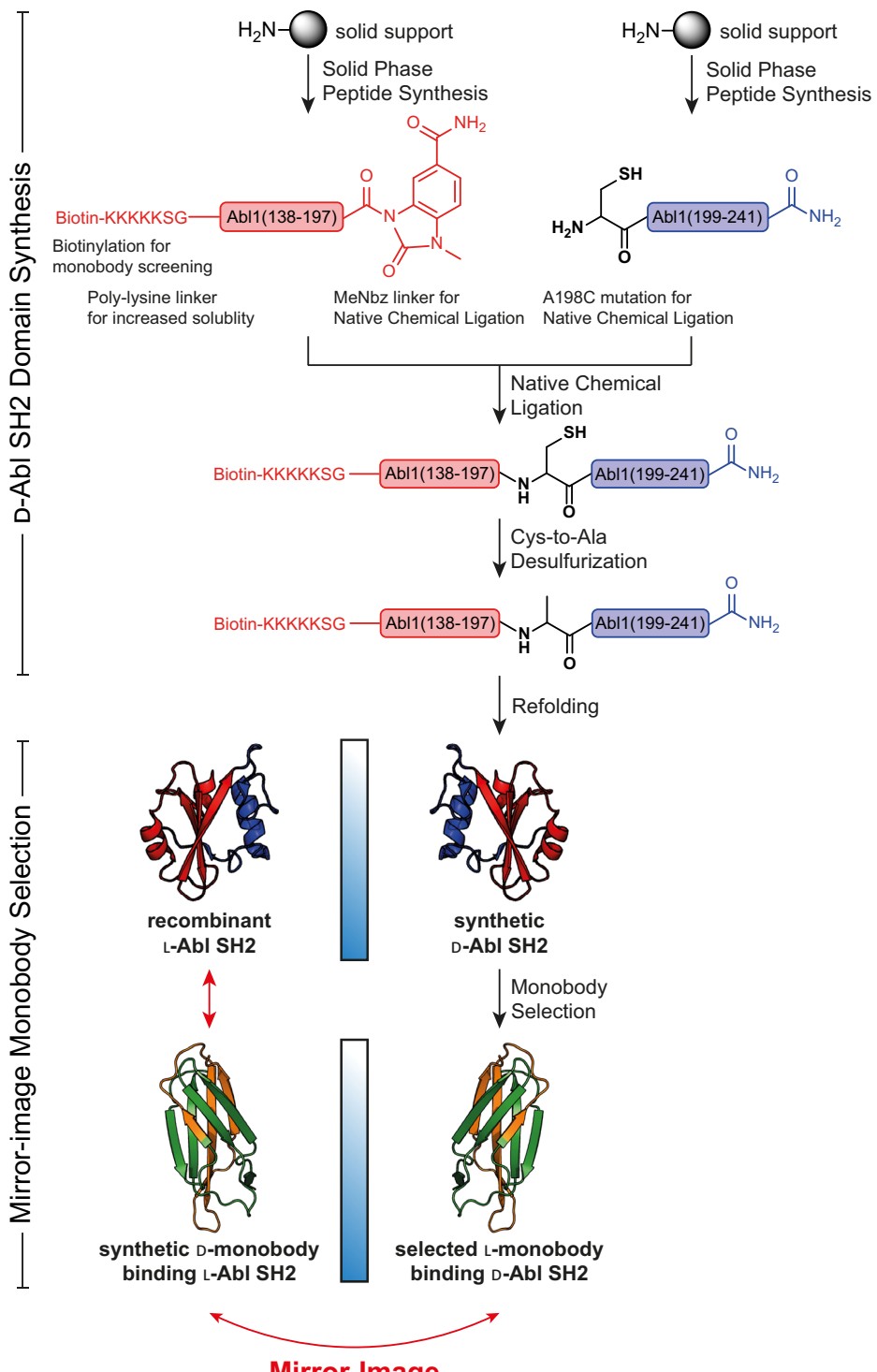

**Fig. 1 | Workflow of Bcr-Abl SH2 synthesis as D-target for mirror-image monobody screening.** The D-Bcr-Abl SH2 domain was prepared by native chemical ligation (NCL) and subsequent desulfurization from two fragments produced by solid-phase peptide synthesis (SPPS), as previously described[34]. After refolding and structure validation, the D-target was subjected to monobody selection through phage and yeast display yielding L-monobody binders. Following binder characterization, most promising monobodies were synthesized in D-configuration resulting in D-monobodies targeting the natural L-Bcr-Abl SH2 domain.

high-affinity binders were identified after four rounds of yeast display with decreasing target concentrations. Six monobody clones with unique sequences were isolated and further characterized (Fig. 2a). Three clones had a diversified CD loop, whereas the others had a CD loop corresponding to the wildtype FN3 sequence[7]. The FG loop, which was identical for four out of the six clones, differed in sequence, but not in length. All clones bound to the D-Abl SH2 domain with similar affinities between 20 and 71 nM, as measured in the yeast display format[13], with clones 21 and 27 (hereafter termed DAM21 and DAM27) showing tightest apparent binding (Fig. 2b). We next expressed and purified the recombinant monobodies in *E. coli* with high yields (44–54 mg/L *E. coli* culture, Fig. 2c, d). Preparative SEC showed high purity and that the elution volumes of all monobodies corresponded to their calculated monomeric molecular weights, except for DAM28,

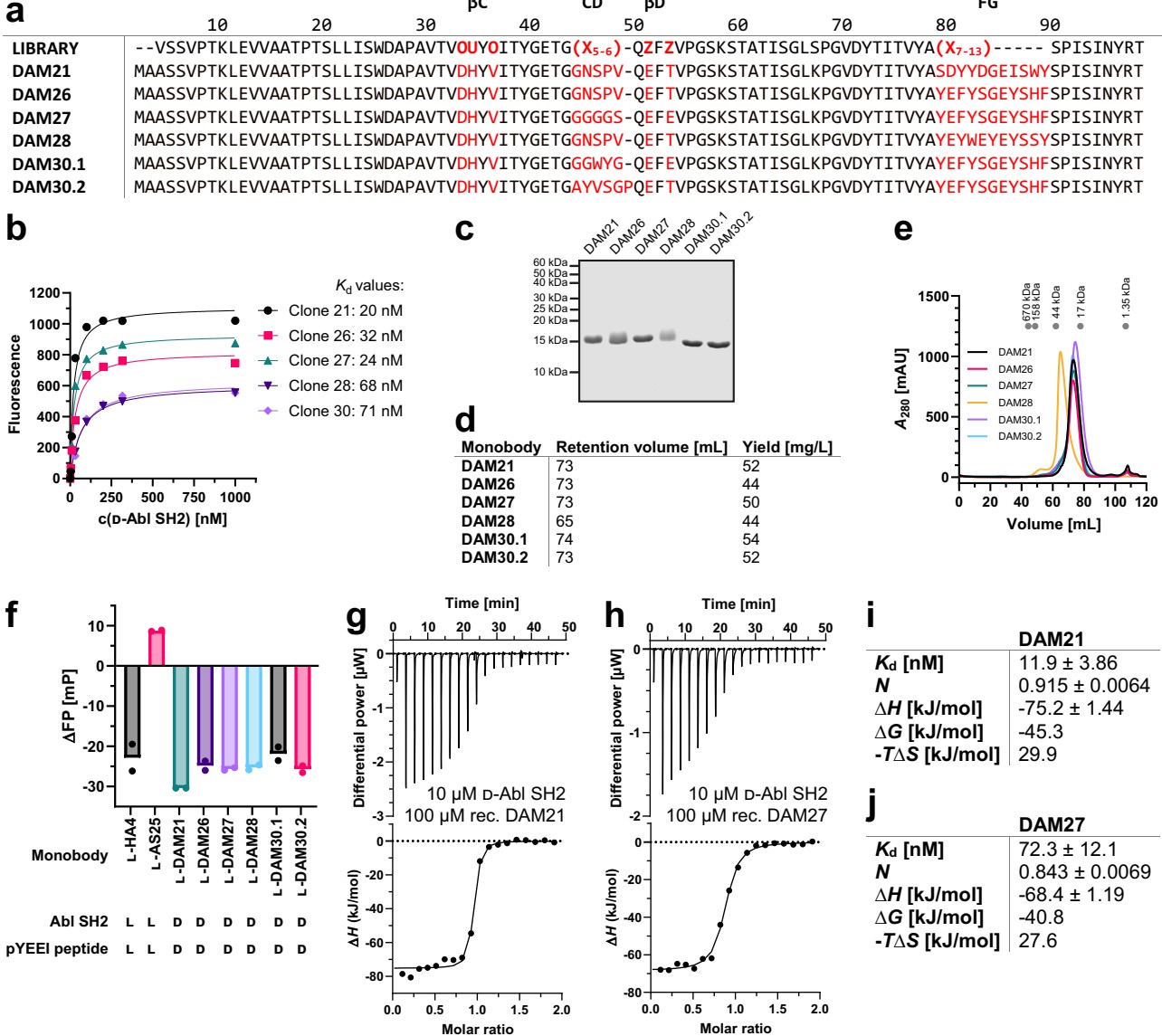

**Fig. 2 | L-monobody selection and characterization against D-Bcr-Abl SH2.**
**a** Amino acid sequences of D-Abl SH2 L-monobody binders (DAM) generated by phage and yeast display selection based on the combinatorial 'side-and-loop' library. In the library designs, "X" denotes a mixture of 30% Tyr, 15% Ser, 10% Gly, 5% Phe, 5% Trp, and 2.5% each of all the other amino acids except for Cys; "O" denotes a mixture of Asn, Asp, His, Ile, Leu, Phe, Tyr, and Val; "U" denotes a mixture of His, Leu, Phe, and Tyr; and "Z" denotes a mixture of Ala, Glu, Lys, and Thr. A hyphen indicates a deletion. **b** Binding titrations in the yeast surface display format to estimate binding affinities of L-monobodies to D-Abl SH2. The mean fluorescence intensities of yeast cells displaying a monobody are plotted as a function of the concentration of the target and fitted to a 1:1 binding model. **c** Sodium dodecyl sulfate-polyacrylamide gel electrophoresis (SDS-PAGE) analysis (performed once), (**d**) retention volumes in size exclusion chromatography (SEC) purification and expression yields, and (**e**) SEC chromatograms of selected L-DAMs recombinantly expressed in *E. coli*. (**f**) Competitive fluorescence polarization (FP) assay of L-DAMs

incubated with D-Abl SH2 and fluorescently labeled D-pYEEI peptide binders in comparison with previously published L-monobodies HA4 (pY peptide competitor) and AS25 (allosteric binder) against L-Abl SH2 incubated with the corresponding L-pYEEI peptide. Measured data from two independent experiments (depicted as dots) were averaged. **g**–**j** Isothermal titration calorimetry (ITC) measurements of recombinant L-monobodies (**g**) DAM21 and (**h**) DAM27 titrated to the synthetic D-Abl SH2 domain. Each panel shows the raw heat signal of an ITC experiment (top) and the integrated calorimetric data of the area of each peak (bottom). The continuous line represents the best fit of the data based on a 1:1 binding model computed from the MicroCal software. Binding parameters including $K_d$ value, stoichiometry ($N$), enthalpy ($\Delta H$), free enthalpy ($\Delta G$) and $-T\Delta S$ calculated from the fit of each experiment are shown in (**i**) and (**j**). A representative measurement of at least two ITC experiments for each monobody is shown. Source data of (**a**, **c** and **f**) are provided as a Source Data file.

which eluted at a higher molecular weight and was therefore not selected for further follow-up (Fig. 2e). These monobodies robustly blocked binding of a D-pY peptide to the D-Abl SH2 domain ($K_d$ = ~5 μM, Fig. S1) as tested with a fluorescence polarization (FP) binding assay. All monobody clones produced a strong FP signal decrease, which was comparable to L-pY competition by the previously characterized L-Abl SH2-targeting monobody HA4[10]. In contrast, monobody AS25 that targets the allosteric SH2-kinase domain interface of ABL1 did not

show competition of the L-pY-SH2 interaction (Fig. 2f)[13]. These results suggest that the DAM monobodies bind to the pY binding site of the SH2 domain. To determine the thermodynamic binding parameters of the monobodies more precisely, we performed isothermal titration calorimetry (ITC) measurements using purified monobody and SH2 proteins. The determined binding affinities in the low to mid nanomolar range were generally consistent with the yeast display measurements (Figs. 2g–j, S2). All measurements suggested a D-Abl

SH2:ʟ-monobody binding stoichiometry of 1:1 and binding enthalpies in the range of −75 to −36 kJ/mol in line with the previously observed strongly enthalpically driven binding of ʟ-monobodies to their ʟ-targets. Given that all monobody clones were pY competitors we focused on DAM21 and DAM27 for further characterization, as they had the highest apparent binding affinities.

### Structures of heterochiral Abl SH2·monobody complexes

To understand the structural basis for monobody recognition of the ᴅ-Abl SH2 domain, we set out to determine the crystal structures of monobody-SH2 complexes. We mixed the synthetic ᴅ-Abl SH2 domain with recombinantly expressed ʟ-DAM21 or ʟ-DAM27 monobody, respectively, and purified the heterochiral 1:1-complexes by SEC (Fig. 3a−e). Of note, to satisfy the large amounts of protein complex needed for crystallization, this batch of the ᴅ-Abl SH2 domain was not desulfurized to alanine after NCL and therefore contained a cysteine residue at position 198 (Table S3, Figs. S3−S5). The alanine to cysteine mutation at this position in recombinant ʟ-Abl SH2 did not change binding affinity to ʟ-monobody AS25 and ʟ-pY peptide (Fig. S6). Crystals were obtained readily or after an additional additive screen. We determined the crystal structures of the ʟ-DAM21-ᴅ-Abl SH2 and ʟ-DAM27-ᴅ-Abl SH2 complexes at 2.7 Å and 2.9 Å resolution, respectively (Fig. 4a, b, Table 1, Fig. S7a, b). In both structures, we found the ᴅ-SH2 domain to be dimerized via a disulfide bridge of Cys198. In addition, we found two different interaction interfaces of the monobody with the SH2 domain, which were very similar in both complexes (Fig. S7c, d). The monobody FG loop, which represents the most extensively diversified region in the library, forms a large part of both interfaces, which have similar buried surface areas (431 Å² and 440 Å² in DAM21, 452 Å² and 427 Å² in DAM27, for interface 1 and 2, respectively). Interface 2 of both structures largely contains the pY peptide-binding interface (346 Å² for the pYEEI peptide, Fig. 4c), whereas interface 1 comprises a region of the SH2 domain surface of unknown

functional significance. Given the robust pY peptide competition of both monobodies (see Fig. 2f) and mutagenesis data described below, we considered interface 2 as the likely biologically relevant interface.

Both monobodies use their long FG loops to present an extended segment that is positioned parallel to the central SH2 β-sheet and therefore perpendicular to the canonical backbone conformation of an SH2 domain-bound pY peptide (Fig. 4a−c). Likewise, several ʟ-monobodies mimic pY peptide binding by also binding perpendicular across the central SH2 β-sheet, which thereby enables access to the pY and +3 pockets that are located on opposite sides of the SH2 β-sheet[12,14]. Unlike those, DAM21/27 form a heterochiral anti-parallel β-sheet between strand Gly84 to Trp88/His88 in DAM21/27 of their FG loop with the βD-strand (Val190 to Arg194) of the ᴅ-Abl SH2 domain (Fig. 4d, e). This interaction corresponds to a "rippled β-sheet", which is a largely neglected structural motif predicted by Pauling and Corey in 1953 and meanwhile experimentally shown to be present in heterochiral peptide structures[47]. In contrast to the canonical "pleated" anti-parallel β-sheets of natural proteins, the side chains experience in rippled β-sheets much less steric crowding. Accordingly, in the DAM21/27 structures, Glu85, the key residue binding to the pY pocket, projects its sidechain in between the SH2 residues His192 and Arg194 in the observed heterochiral rippled β-sheet, whereas it would sterically clash with these residues in a pleated β-sheet, due to the different register of β-sheet hydrogen bonds (Fig. 4d, e).

The rippled β-sheet position Glu85 in the FG loop is inserted into the SH2 pY pocket, where it forms an ionic interaction with SH2 residue Arg171, which is part of the conserved FLVRES motif and makes the most critical interaction with the phosphate group in pY peptide ligands (Fig. 4f, g)[48]. In contrast to DAM21/27, most other SH2-targeting ʟ-monobodies that engaged the pY pocket rather used tyrosine residues in the FG loop and often a phosphate or sulfate from the crystallization buffer to mimic pY[10,14]. In DAM21, FG loop residue Asp83 forms an additional ionic interaction with Arg194, unique in the Abl1

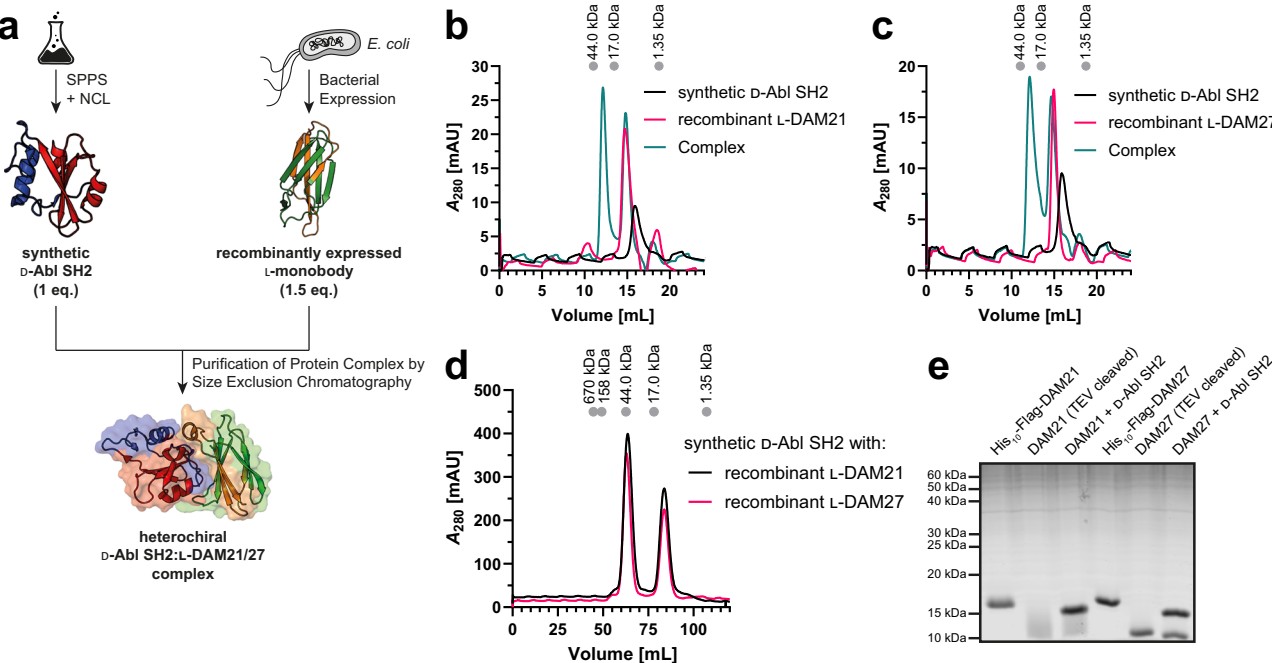

**Fig. 3 | Complex formation and structure elucidation of the heterochiral ᴅ-Abl SH2:ʟ-monobody interaction. a** Workflow for generation of the heterochiral ᴅ-Abl SH2:ʟ-monobody complex. ᴅ-Abl SH2 was produced by solid-phase peptide synthesis (SPPS) and native chemical ligation (NCL) and the ʟ-monobody by bacterial expression and subsequent purification. After mixing, the complex was purified by size exclusion chromatography (SEC). **b, c** Analytical SEC of complex formation between synthetic ᴅ-Abl SH2 and recombinantly expressed (**b**) ʟ-DAM21 and (**c**) ʟ-DAM27. **d** Preparative SEC of ᴅ-Abl SH2:ʟ-monobody complexes generated for crystallization condition screening. All chromatograms include a calibration of the column with standard proteins on top. **e** Sodium dodecyl sulfate-polyacrylamide gel electrophoresis (SDS-PAGE) analysis of the complex formation (performed once). Source data of (**e**) are provided as a Source Data file.

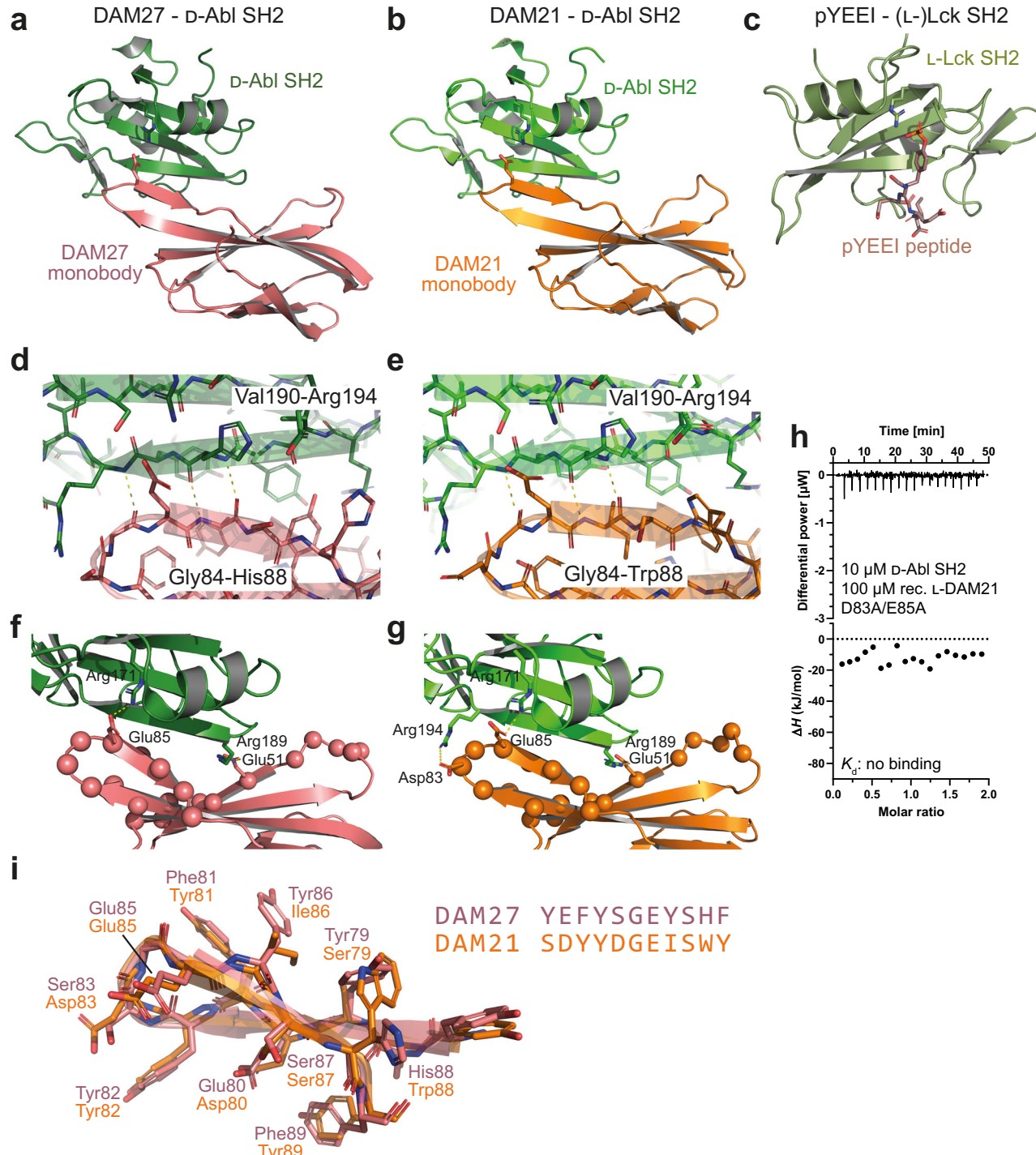

**Fig. 4 | Crystal structures of heterochiral monobody-Abl SH2 complexes.**
**a**, **b** Ribbon representation of the ᴅ-Abl SH2 domain structure (green) bound to the (**a**) DAM27 or (**b**) DAM21 monobodies in ʟ-configuration. Arg171 of the SH2 FLVRES motif, which is critical for pY binding, as well as Glu85 in the FG loop of the monobodies is shown in stick representation. **c** Structure of an (ʟ-) phosphotyrosine peptide (pYEEI, stick representation) bound to the (ʟ-)SH2 domain of the Src family kinase Lck (PDB: 1LKK). The critical Arg residue of the FLVRES motif in the pY binding pocket is shown in stick representation. **d**, **e** Close-up view of the ᴅ-SH2:(ʟ-) monobody interface (DAM27, panel **d**; DAM21, panel **e**) highlighting the heterochiral "rippled" anti-parallel β-sheet between strand Gly84 to Trp88/His88 in DAM21/27 of the monobody FG loop with Val190 to Arg194 of the βD-strand of the ᴅ-Abl SH2 domain. **f**, **g** Close-up view of the ᴅ-SH2:(ʟ-)monobody interface (DAM27, panel **f**; DAM21, panel **g**). The ionic interactions discussed in the "Results" section are shown as sticks and are labeled. The position of randomized residues in the monobody side-and-loop library is shown as balls. **h** Isothermal titration calorimetry (ITC) measurements of recombinant ʟ-DAM21 D83A/E85A mutant titrated to the synthetic ᴅ-Abl SH2 domain. The panel shows the raw heat signal of an ITC experiment (top) and the integrated calorimetric data of the area of each peak (bottom). **i** Structural superposition and sequence alignment of the FG loops of the DAM27 and DAM21 monobodies.

SH2 domain and therefore a possible contributor to selective Abl1 binding (Fig. 4g). In DAM27, this residue is replaced with a serine. To probe the functional roles of Asp83 and Glu85, we produced a D83A/E85A mutant in DAM21. The mutant protein displayed an almost identical far-UV circular dichroism spectrum, and thus similar secondary structure content, as wildtype DAM21 (Fig. S8, Table S4). ITC measurements showed no detectable binding (Fig. 4h), thus validating the critical role of these residues for SH2 domain binding and interface 2 as the biologically relevant protein interaction site. This is noteworthy, as glutamate is not a suitable pY mimetic in SH2 ligand peptides.

An additional ionic interaction was observed in the DAM21/27-SH2 interface: SH2 residue Arg189 interacts with Glu51 in the ßD strand of the monobody and is located at the other end of the heterochiral SH2-monobody interface (Fig. 4f, g). Interestingly, in the monobody side-and-loop library, this position in the ßD strand is a mixture of Ala, Glu, Lys, Thr, of which only Glu is suitable to interact with Arg189. Taken together, the DAM21/27 interfaces are much more prominent in ionic interactions when compared to other (homochiral) monobody-SH2 interfaces.

Structural superposition of the FG loop conformations of DAM27 and DAM21 revealed a strikingly similar conformation despite their sequence differences in seven out of eleven amino acid positions (Fig. 4i). This can be explained by a pairwise compensatory exchange of bulky and less bulky hydrophobic amino acids. This is particularly notable at the first and second to last position. In DAM27, Tyr79 packs towards the β-hairpin and His88 is solvent exposed, whereas in DAM21, Trp88 is rotated inwards and would clash with large hydrophobic or aromatic residues at this position. Hence, Tyr79 is exchanged to Ser79 in DAM21 (Fig. 4i).

In summary, monobody recognition is dominated by the FG loop of both monobodies that form a rippled β-sheet with the βD-strand of the SH2 domain, as well as ionic interactions between glutamate residues of the monobody and arginine residues of the SH2 domain.

## Synthesis of D-monobody DAM27

Since the monobody scaffold lacks endogenous cysteines, we planned a two-segment synthesis for DAM27 by SPPS with Thr60-Ala61 as the ligation junction through NCL and subsequent desulfurization resulting in the N- and C-terminal monobody fragments DAM27(4-60) and DAM27(61-98), respectively, spanning the entire length of the FN3 domain (Fig. 5a). In the C-terminal peptide, the N-terminal alanine was mutated to cysteine as required for NCL. Synthesis was first optimized in L-configuration due to lower reagent costs and subsequently applied for D-monobody synthesis. Both DAM27 N- and C-peptides could be synthesized without major need for optimization and obtained with high purity after HPLC purification (Table S5, Figs. S9–S12). We used a similar strategy to prepare the two peptides for NCL, as previously optimized for the synthesis of the D-Abl SH2 domain including the usage of the 3-amino-4-(methylamino)benzoic acid (MeDbz) linker and N-terminal capping strategy[34]. However, the product after NCL was insoluble and hence could not be analyzed and purified.

To improve the solubility of monobody fragments, we included three repeats of the highly soluble XTEN peptide (six amino acids) at the C-terminus of the C-terminal DAM27 peptide[49]. Gratifyingly, the product after NCL was soluble and obtained with ~20% yield after HPLC purification in both L- and D-configuration (Figs. S13, S14). Subsequent desulfurization using our optimized protocol[34] was not successful for the DAM27 ligation product, but an alternative protocol using 2-mercaptoethane sulfonate (MESNa) instead of glutathione resulted in quantitative conversion of cysteine to alanine giving rise to the native DAM27 monobody sequence (Figs. S15, S16). Refolding of the DAM27 protein was achieved by dialysis into aqueous buffer directly using the desulfurization reaction without a need of HPLC purification due to complete conversion (Figs. S15, S16). The protein was purified by SEC to remove aggregates and to determine oligomerization state. DAM27 both in L- and D-configuration eluted in a predominantly monomeric peak in line with the recombinantly expressed DAM27 control (Fig. 5b, c) and overall yield was ~1%. In summary, a facile synthesis strategy for the DAM27 monobody from two peptide segments could be established, which supplied sufficient protein for further biophysical and functional characterization.

## Synthesis of the split-D-monobody DAM21

After establishing the conventional synthesis of DAM27 from two pieces including NCL, we aimed at testing a more rapid and versatile approach to obtain D-monobodies without the need to optimize synthesis conditions. Inspired by early work that showed complementation of a functional monobody by split fragments in a yeast-surface-two-hybrid system[50], we attempted to obtain a functional D-monobody by mixing N- and C-terminal fragments without NCL. As observed in the DAM21 crystal structure, the monobody CD loop is not involved in Abl SH2 binding. Therefore, we decided to split the sequence between Asn45 and Ser46 in the CD loop generating the fragments DAM21(4-45) and DAM21(46-98) (Fig. 5d). Luckily, these fragments were previously shown to be the most efficient in complementing a functional L-monobody[50]. Both DAM21 N- and C-peptides in L- and D-configuration could be synthesized without need for major optimization (Table S5, Figs. S17–S20). Although the C-terminal peptide was initially aggregating during HPLC purification, this peptide could be obtained employing an HPLC-free purification strategy, in which the desired peptide is linked to agarose beads and acetylated, truncated peptides are washed off, as they are not reacting with the beads (Figs. S19, S20). Both peptides were then mixed in a 1:1 molar ratio and refolded by dialysis. SEC purification showed a predominantly monomeric protein, but in contrast to DAM27 an appreciable peak close to the void volume was observed indicating formation of some higher-order oligomers or aggregates during refolding (Fig. 5e). In summary, we were able to generate a monomeric split version of DAM21 from two peptide fragments (Fig. 5f, g) for further biophysical and functional characterization.

## Folding and stability of synthetic DAM21 and DAM27

We next analyzed purity, folding and stability of our synthetic L- and D-DAM21/27. Gel electrophoresis analysis revealed that the synthetic proteins are migrating at the expected molecular weight without detectable impurities (Fig. 6a, b). Of note, recombinant L-DAM27-XTEN migrated slightly higher than synthetic L- and D-DAM27-XTEN, because the recombinant construct carries four additional N-terminal amino acids from cloning and both synthetic constructs contain biotin. For small proteins, such as monobodies, these seemingly minor differences can result in a detectable difference in migration distance (Fig. 6b). Far-UV circular dichroism spectroscopy showed very similar spectra in terms of shape, signal strength and depth of minima for recombinant DAM27 and synthetic L-DAM27 both containing the XTEN extension (Fig. 6c, Table S6), in line with a pure β-sheet protein. The lower mean residue ellipticity (MRE) below 210 nm as compared to recombinant DAM27 lacking XTEN is in line with the XTEN peptide being unstructured. The synthetic D-DAM27 showed a mirrored spectrum at the x-axis (Fig. 6c). Secondary structure predictions with BeStSel[51] revealed almost identical content of β-sheets and β-turns, and absence of α-helices, between recombinant and synthetic L-/D-DAM27 versions (Table S6). For the split-L-DAM21 monobody, a spectrum in line with a β-sheet protein was obtained although with some spectral differences to the recombinant (non-split) DAM21 below 215 nm (Fig. 6d, Table S7). As for split-D-DAM21 the spectrum is mirrored along the x-axis (Fig. 6d). Despite these spectral differences, secondary structure predictions still showed a similar secondary

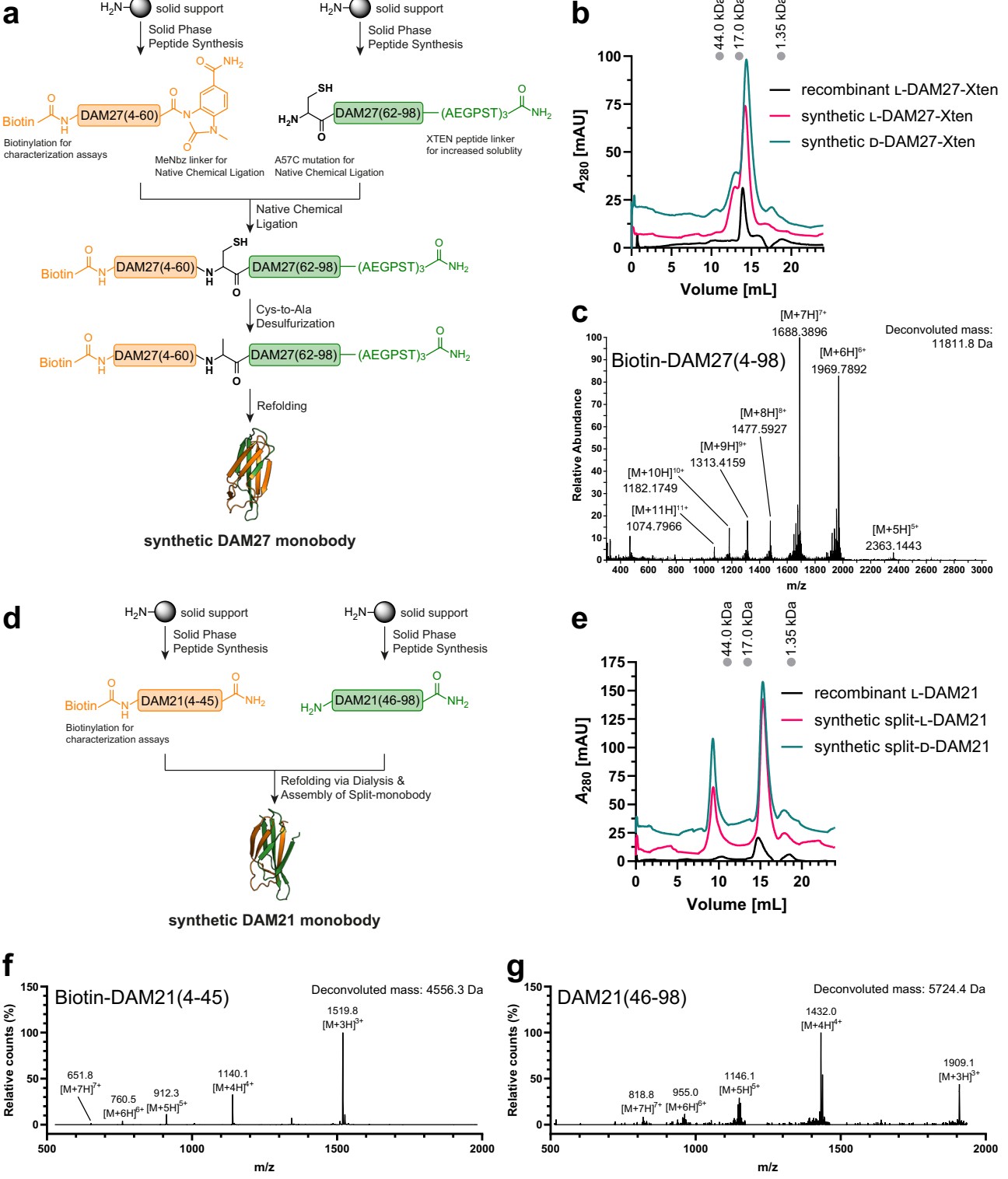

**Fig. 5 | DAM21 and DAM27 synthesis, refolding and purification. a** Strategy of DAM27 synthesis. The N- and C-terminal fragments are shown in orange and green, respectively, and synthesized via solid-phase peptide synthesis (SPPS). The N-terminal peptide corresponds to the biotinylated DAM27(4-60) fragment with C-terminal MeDbz linker, which is activated to MeNbz on resin, and the C-terminal peptide resembles DAM27(62-98) with an N-terminal cysteine (Cys61) residue. After cleavage from the resin, both peptides can undergo native chemical ligation (NCL) with subsequent desulfurization to yield full-length monobody DAM27(4-98). MeDbz: 3-amino-4-(methylamino)benzoic acid; MeNbz: N-acyl-N-methylacylurea. **b** Size exclusion chromatography (SEC) and (**c**) high-performance liquid chromatography-mass spectrometry (HPLC-MS) analysis of the final refolded L- and D-DAM27 proteins in comparison with the recombinantly expressed L-DAM27-XTEN reveals similar SEC retention volumes and expected masses of the synthetic proteins. **d** Strategy of DAM21 synthesis. The N- and C-terminal peptides in orange and green obtained by SPPS represent the biotinylated DAM21(4-45) and DAM21(46-98) fragments, respectively. After mixing of the peptides and refolding by dialysis, the full-length split-monobody DAM21(4-98) is obtained. **e** SEC of the final refolded L- and D-DAM21 proteins in comparison with the recombinantly expressed L-DAM21 reveals similar SEC retention volumes of the synthetic proteins. **f, g** HPLC-MS analysis of the (**f**) DAM21(4-45) and (**g**) DAM21(46-98) fragments.

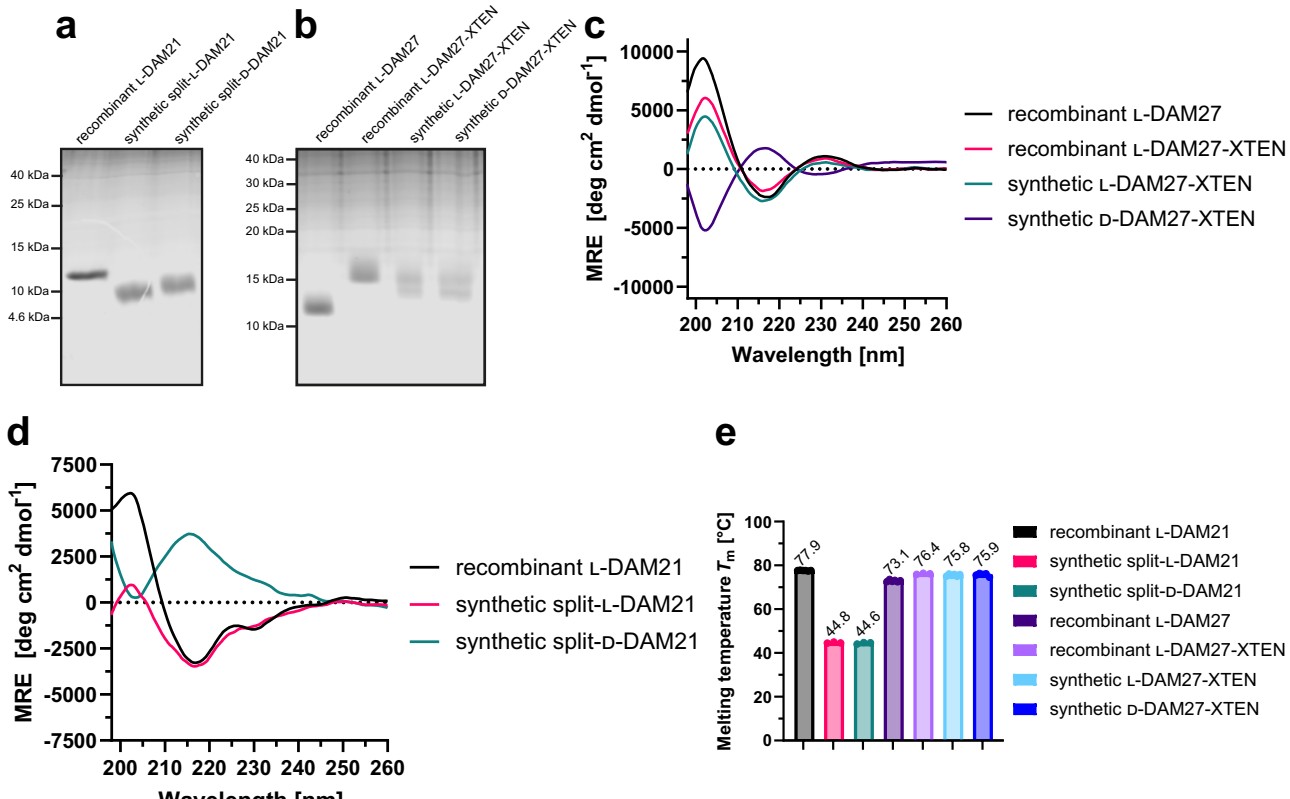

**Fig. 6 | Folding and stability characterization of synthetic L- and D-monobodies.**
**a, b** Representative analysis of purity of synthetic (**a**) L- and D-DAM21 and (**b**) L- and D-DAM27 by sodium dodecyl sulfate-polyacrylamide gel electrophoresis (SDS-PAGE) (performed at least twice). **c, d** Averaged far-UV circular dichroism (CD) spectra from three independent measurements of synthetic (**c**) L- and D-DAM27 and (**d**) L- and D-DAM21 in comparison with their recombinantly expressed analogues.

Mean residue ellipticity (MRE) was calculated from three independent measurements. **e** Bar graph representation of thermal stability of recombinantly expressed and synthetic L- and D-monobodies assessed by nano differential scanning fluorimetry (nanoDSF). Melting temperatures ($T_m$) were measured at least three times (depicted as dots) and averaged. Error bars represent the standard deviation (SD). Source data of (**a**, **b** and **e**) are provided as a Source Data file.

structure content especially between recombinant L-DAM21 and split-D-DAM21 (Table S7). Next, we measured thermal stability by nano differential scanning fluorimetry (nanoDSF). For DAM27 identical melting temperatures around 75 °C were observed for the synthetic L- and D-versions and its recombinant analogue (Fig. 6e, Fig. S21a, c). The presence of the XTEN peptide resulted in a mild stabilization by ~3 °C (Figs. 6e, S21a, c). Interestingly, unfolding was fully reversible for all proteins (Fig. S21b, d). As expected for a split protein, synthetic split-L- and D-DAM21 showed a lower melting temperature ($T_m$) around 45 °C as compared to the recombinant (non-split) version (-78 °C) (Figs. 6e, S21e, g). Despite this, thermal unfolding was reversible as for DAM27 (Fig. S21f, h). In summary, both synthetic strategies resulted in folded monobodies.

**Protease and plasma stability of DAM27**
We next assessed protease stability of DAM27 by incubation with two different broad-spectrum proteases, pepsin and proteinase K, and analyzed the digests after 4 and 24 h of incubation. Both L-versions of the recombinant and synthetic DAM27 showed complete degradation after 24 h. Minor residual protein was detected after 4 h of pepsin digestion (Fig. 7a, b). In contrast, D-DAM27 was completely resistant to degradation by both proteases. Sufficient plasma stability is a crucial prerequisite for the envisioned use of D-monobodies as protein therapeutics. After incubation of L-DAM27 in mouse plasma for up to 96 h, we observed a strong reduction of L-DAM27 signal at 24 h and all later time points. In contrast, D-DAM27 showed no signal reduction for up to 96 h (Fig. 7c). These results show very high protease resistance and plasma stability of D-monobodies.

**D-DAM21 and D-DAM27 bind to native L-Bcr-Abl SH2**
After confirming folding and stability of the D-monobodies, target binding was measured by ITC. For DAM27 binding affinities between 100 and 128 nM were measured for the interaction of the D-monobody with the L-target and vice versa (Fig. 8a–c). Of note, in the absence of the XTEN peptide a slightly higher affinity was measured (Fig. 2h, j). For DAM21 both split-L- and D-versions showed an interaction of around 100 nM with D- and L-Bcr-Abl SH2, respectively (Fig. 8d, e). While this affinity is in the same range of the D-DAM27:L-Bcr-Abl SH2 interaction, it is around 10-fold lower than the recombinant (non-split) DAM21 with D-Abl SH2 (Fig. 2g, i). All measurements showed a 1:1 binding stoichiometry and negative binding enthalpies (Fig. 8a–e). As expected, no binding was detected for L-DAM21 and L-DAM27, respectively, to the L-Bcr-Abl SH2 domain (Fig. S22).

**Inhibition of Bcr-Abl kinase activity by D-monobodies**
After showing binding to the Bcr-Abl SH2 domain with high affinity, we next wanted to assess a possible inhibitory effect of monobodies on Bcr-Abl kinase activity in vitro. We recombinantly expressed and purified BCR::ABL1 fragments either containing the kinase domain (KD) alone or a larger fragment also containing the SH2 domain (SH2-KD)[52]. After addition of different monobody proteins, kinase activity was measured using a radiometric kinase assay[13]. Besides the L- and D-versions of DAM21 and DAM27, we included the previously characterized Abl SH2 binding monobodies HA4 and AS25 as well as the non-binding control monobody HA4 Y87A[10,13]. In vitro kinase activity of Abl1 KD (not including the Abl1 SH2 domain) remained unchanged upon incubation with all monobodies indicating no unspecific

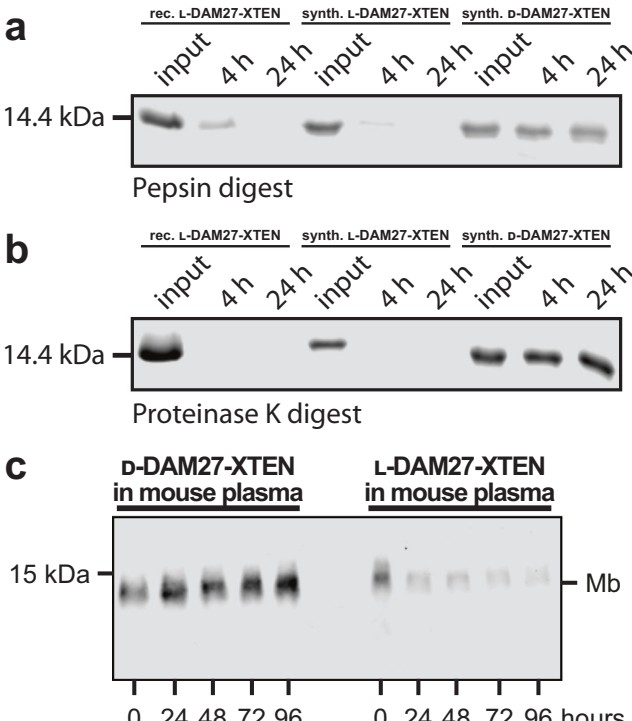

**Fig. 7 | Protease resistance and mouse plasma stability of synthetic L- and D-monobodies. a, b** Protease resistance of recombinant L-DAM27-XTEN as well as synthetic L- and D-DAM27-XTEN after incubation with (**a**) pepsin and (**b**) proteinase K was analyzed by sodium dodecyl sulfate-polyacrylamide gel electrophoresis (SDS-PAGE). **c** Plasma stability of synthetic L- and D-DAM27-XTEN. Monobodies labeled with biotin at the N-terminus after incubation with mouse plasma were analyzed by Western blotting and detection of biotin using Streptavidin-IRDye680. A representative blot of three repeats is shown. Source data of (**a–c**) are provided as a Source Data file.

inhibitory or activating effect of the monobodies on Abl1 kinase activity (Fig. 8f). In contrast, D-DAM27 resulted in a major reduction of kinase activity of SH2-KD, comparable to inhibition by the validated allosteric Bcr-Abl inhibiting monobody AS25. In contrast, no inhibition was observed with L-DAM27. Also, no inhibition was observed with HA4 and HA4 Y87A, in line with previous observations (Fig. 8g)[10]. Although we noted a minor and unexplained increase in kinase activity by split-monobody L-DAM21, D-DAM21 showed significant inhibition as compared to its L-counterpart, although to a lesser extent than D-DAM27 (Fig. 8g). These data show that D-monobodies binding to the Abl1 SH2 domain are able to inhibit the kinase activity of Bcr-Abl, which is crucial for BCR::ABL1 signaling and CML maintenance.

### High selectivity of D-monobodies for the BCR::ABL1 SH2 domain

Next, we studied selectivity of the D-monobodies for the BCR::ABL1 SH2 domain. Analysis of the DAM21/27 crystal structures (see Fig. 4a, b) indicated that the CD loop of the Abl SH2 domain is positioned close to the βC and βD strands of DAM27 and DAM21 (Fig. 9a). The Abl1 (and Abl2) SH2 domain has a particularly short CD loop, which seems to be a requirement for binding to DAM21 and DAM27 (Fig. 9a). In contrast, all SH2s of the Src- and Tec-family (9 and 5 members, respectively), the closest paralogues of the Abl family, have CD loops that are four to six amino acids longer. Molecular modeling indicates that these longer CD loops would clash with the βC/βD strands of DAM21 and DAM27 (Fig. 9b, c). Also, several other SH2 families including the Jak kinases, STAT transcription factors and SHP1/2 (C-SH2) tyrosine phosphatases have such long CD loops (Table S8). In addition, the ionic interaction of Glu51 and Arg189 (Fig. 4f, g), can only be formed with Abl kinase SH2

domains, as many other SH2 domains, including the Src kinase SH2s, do not have Arg (or Lys) in this position (Fig. 9c, Table S8). To assess this predicted selectivity experimentally, ITC measurements with the SH2 domains of Bruton's tyrosine kinase (Btk) and lymphocyte-specific protein tyrosine kinase (Lck), as representatives of the closely related Tec- and Src-family showed that both D-DAM27 and D-DAM21 did not bind to these SH2 domains (Fig. 9d–g). Overall, these observations indicate a high selectivity of DAM21 and DAM27 for the Abl SH2 domain.

### Binding of D-monobodies to Bcr-Abl in cell lysates and permeabilized CML cells

In order to determine if D-monobodies can bind to full-length BCR::ABL1 in the context of the complex proteome of human cells, we performed pulldown experiments with the biotinylated L- and D-DAM21/27 variants in three biological repeats from lysates of the BCR::ABL1-expressing cell line K562, one of the most commonly used cell lines in CML research, followed by quantitative proteomics analysis of bound proteins. This approach also enables to complement the findings on D-monobody selectivity described above and to determine its interactome. Comparison of D- vs. L-DAM21 showed higher abundance of BCR and ABL1 in the samples from the D-DAM21 pulldown when compared to its respective L-counterpart (Fig. 9h, Table S9). Comparing the pulldowns of D- vs. L-DAM27 showed higher abundance of BCR, but only a mild enrichment of ABL1, which was not significant (Fig. 9i, Table S10). Since BCR::ABL1 is a fusion oncoprotein, enrichment of either BCR or ABL1 is sufficient to show binding of D-DAM27 to BCR::ABL1. Interestingly, we also found several previously identified and validated BCR::ABL1 interactors, including kinases (CDK1, MTOR, PRKDC)[53–55], proteins involved in transcription regulation, splicing and cell proliferation control (STAT1, FUS, GTF3C4, YTHDC1, EPS15)[56–61], proteins mediating proteasomal degradation (CUL4B, TRIM25, PSMA4)[62–64], transport proteins and chaperones (AP2A1, XPO1, HSPD1)[61,65–67], as well as cytoskeletal components (EMD, SPTA1)[68,69], significantly enriched with the two D-monobodies (Fig. 9h, i, Tables S9, S10). In line with previous observations with other Abl SH2-targeting monobodies[10], it is likely that these proteins piggyback on BCR::ABL1 and are therefore not direct interactors of the monobodies. Importantly, no other SH2 domain-containing proteins, such as BTK, CSK, SH2B1 and TYK2, were significantly enriched with the D-monobody variants, despite the expression of ~70 SH2 domain-containing proteins in K562 cells, and thus confirming the ITC measurements with the Btk SH2 domain resulting in no binding (Fig. 9d, f). As expected for single-step pulldown mass spectrometry experiments, the dataset contains a lot of common background contaminants, which are listed in the Contaminant Repository for Affinity Purification (CRAPome)[70] and were therefore disregarded. Together, the proteomics data show preferential binding of D-DAM21 and D-DAM27 to BCR::ABL1 complexes over their L-counterparts and that both D-monobodies are able to bind BCR::ABL1, but no other SH2-containing proteins in the complex proteome of a human cell.

In addition, we used a FACS-based assay, in which we permeabilized K562 cells and monitored the binding of the D-monobodies by a shift in fluorescence signal. Without optimization of assay parameters, binding of D-DAM21 and D-DAM27 was readily observed to be much stronger than the binding of synthetic L-DAM21 and L-DAM27 and exceeded binding of the allosteric BCR::ABL1 binder/inhibitor AS25 (Fig. S23). No binding was observed for the negative control monobody HA4 Y87A (Fig. S23).

These two assays show binding of the two D-monobodies to full-length BCR::ABL1 in cell lysates and permeabilized human BCR::ABL1-expressing cells.

### Discussion

We and a parallel study by Hayashi et al.[46] show that D-monobodies for two therapeutically validated targets with different fold, sequence and

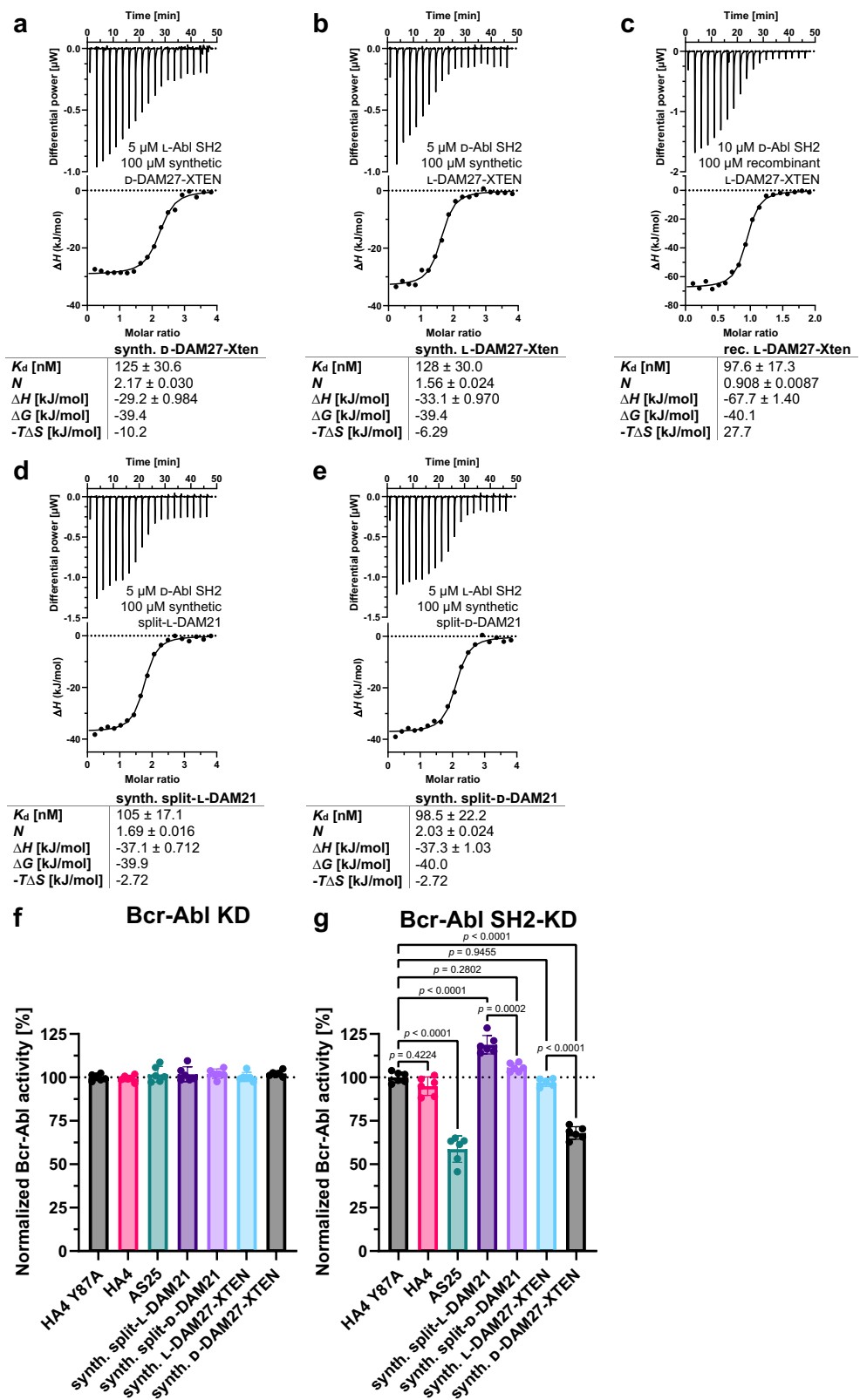

function can be generated. Some main technical differences of the two studies are the use of monobody libraries with different positions and degrees of residue randomization and different selection techniques: Phage and yeast display in this study and mRNA display in the Hayashi et al. paper. Importantly, also different strategies were used to chemically synthesize D-monobodies: A 2-piece NCL strategy and assembly of a split-D-monobody was used in our study, whereas Hayashi et al. employed a 3-piece NCL strategy. Despite these strong differences,

both studies converge to a common endpoint, the development of a functional high-affinity D-monobody. This convergence argues for the robustness of the workflows and promise that these can be applied to develop mirror-image monobodies for a broad range of additional targets.

Monobodies hit a sweet spot for the development of D-protein therapeutics, as they can be developed to bind diverse paratope topologies, which enables a broad spectrum of possible targets[8].

**Fig. 8 | Binding of synthetic L- and D-monobodies to D- and L-Bcr-Abl SH2.**
**a**–**e** Isothermal titration calorimetry (ITC) measurements of (**a**) synthetic D-DAM27-XTEN titrated to recombinantly expressed L-Abl SH2, (**b**) synthetic L-DAM27-XTEN and (**c**) recombinant L-DAM27-XTEN both titrated to synthetic D-Abl SH2, (**d**) synthetic split-L-DAM21 and (**e**) synthetic split-D-DAM21 titrated to synthetic D- and L-Abl SH2, respectively. Each panel shows the raw heat signal of an ITC experiment (top) and the integrated calorimetric data of the area of each peak (bottom). The continuous line represents the best fit of the data based on a 1:1 binding model computed from the MicroCal software. Binding parameters including $K_d$ value, stoichiometry ($N$), enthalpy ($\Delta H$), free enthalpy ($\Delta G$) and $-T\Delta S$ calculated from the fit of each experiment are shown below. A representative measurement of at least two ITC experiments for each monobody is shown. **f**, **g** Measurement of kinase activity of (**f**) Bcr-Abl kinase domain (KD) and (**g**) SH2-KD after incubation with synthetic split-L- and D-DAM21 as well as synthetic L- and D-DAM27-XTEN in comparison with binding control monobodies HA4 and AS25 and the non-binding control monobody HA4 Y87A using a radiometric kinase assay. All monobodies were used at a concentration of 5 μM. Here, radioactively labeled $^{32}$P was incorporated into a biotinylated substrate peptide by recombinantly expressed KD and SH2-KD and detected via scintillation counting. Six independent experiments were performed (depicted as dots) and averaged. Error bars represent the standard deviation (SD) and statistical analysis was done with a one-way ANOVA and Sidak's test. The calculated $p$-values are depicted in (**g**) and were considered statistically significant below a value of 0.05. $F$ values and degrees of freedom were 124.3 and 40. Source data of (**f**, **g**) are provided as a Source Data file.

Furthermore, the small size of monobodies promises a more straightforward chemical synthesis as compared to larger synthetic binders, such as DARPins, repebodies or anticalins or binder classes that have disulfide bridges.

Our study also illustrates an alternative way of how an SH2 domain can be targeted by heterochiral protein-protein interactions that are distinct from previously developed monobodies. Including the co-crystal structures presented here, we have solved 8 structures of SH2 domain-monobody complexes in which the monobody inhibits pY ligand binding[10,12,14]. Collectively, five different binding/inhibition modes can be distinguished: (1) The FG loops of monobodies HA4 and Mb(Yes_1) targeting the Abl1- and Yes-SH2, respectively, carry a tyrosine residue, which, together with an inorganic sulfate or phosphate ion, respectively, closely mimics a pY peptide ligand[10,14]; (2) Monobodies Mb(Lck_1) and Mb(Lck_3) bind the Lck SH2 domain and block the +3 specificity pocket with their respective CD loop[14]; (3) and (4) Monobodies NSa1 and CS1 show a pY-independent mode of interaction with the N- and C-terminal SH2 domain of SHP2, respectively. They still mimic the bound peptide and bind the same epitope, but bind in two opposite orientations[10,12,14]; (5) The DAM21/27 monobodies described in this study engage the pY pocket using a glutamate residue in the FG loop, several salt bridges and a heterochiral rippled β-sheet that runs perpendicular to the peptide-binding groove and hence drastically distinct to all other binding modes. Hence, one may envision that D-monobodies could increase the number of accessible binding modes for this and other L-targets. Overall, the five different SH2 binding modes and mechanisms of pY competition that were observed so far may be one reason for the high selectivity of SH2-targeting monobodies.

The recent rise of flow chemistry has enabled the production of peptide chains of up to 164 amino acids in one step and will facilitate D-protein synthesis[71]. On the other hand, our 2-piece D-monobody synthesis strategy has the advantage that the C-terminal peptide contains the FG loop and the randomized library residues of the βC/βD strands, both major contributors to monobody-target binding, whereas the N-terminal peptide only contains that CD loop, which often is not involved in target binding. Hence, we envisage reusing the N-peptides and only swapping the C-peptide for the synthesis of other continuous and split-D-monobodies without the need for further protocol modifications. We also demonstrated that split-D-monobodies can be made by 1:1 mixing of N- and C-terminal monobody peptides. Advantages of this strategy include a more facile and quicker development, as no NCL needs to be optimized and circumventing a possible insoluble ligation product, saving a purification step and making desulfurization unnecessary. Also, there is no need to change the native sequence, as the split site, in contrast to NCL does not require a particular sequence. On the other hand, the CD loop cannot be used as the preferential split, if it is involved in target binding, which might be hard to evaluate in the absence of structural information or mutagenesis experiments. In addition, while high-affinity binding is retained, the split monobody had a reduced binding affinity and also some impact on folding and stability was observed.

The next steps that go beyond the scope of this paper are to determine pharmacokinetics and biodistribution of D-monobodies in vivo, to incorporate suitable intracellular delivery approaches and to evaluate a possible superior intracellular stability of D-monobodies. Our lab is currently finishing work on cellular delivery using recombinant "supercharged" (L-)monobodies and hence will be the focus of future research.

While there are enormous efforts and progress to facilitate development strategies for D-protein binders, a clear bottleneck is the necessity to synthesize both target protein and binder in D-configuration, as combinatorial libraries for phage, yeast or mRNA display selection are only available in L-configuration. Therefore, efforts to replicate biological systems in D-configuration are important developments to eventually construct a self-replicating and -expressing mirror-image biological system. Advances include the synthesis of mirror-image DNA polymerases Dpo4[72] and Pfu[73], DNA ligase[74], T7 RNA polymerase[75] and 5S ribonucleoprotein complexes as components of a functional mirror-image ribosome[76]. Additionally, D-proteins can be sequenced by mirror-image trypsin digestion[77].

## Methods

### Antibodies, cell lines and reagents
The following antibodies, secondary antibodies and fluorophore-coupled streptavidin were used for monobody selection and yeast display binding assays: mouse anti-V5 (Thermo Fisher Scientific, MA5-15253) and FITC-coupled anti-mouse IgG (Sigma-Aldrich, F0257-5ML), streptavidin-DyLight650 (Thermo Fisher Scientific, 84547). AlexaFluor488-coupled Streptavidin (S32354) used for detection of monobodies binding to Bcr-Abl in cells was purchased from Thermo Fisher Scientific. For Western blots of plasma stability assays IRDye 680-Streptavidin (926-68079) was purchased from LiCOR and used at 1:10,000 dilution in 5% Milk in Phosphate-Buffered Saline (PBS-T). Non-sterile mouse plasma with sodium heparin (ABIN925342) was purchased from antibodies-online/Rockland Immunochemicals. Streptavidin MagneSphere Paramagnetic Particles (Z5481, Promega) were used during monobody selection. K562 cells were purchased from DSMZ (ACC-10, Deutsche Sammlung von Mikroorganismen und Zellkulturen, Braunschweig, Germany). Further reagents used are listed in the Supplementary Information.

### Monobody selection
Monobodies were selected according to methods previously described using biotinylated target proteins[7,78]. Briefly, four rounds of phage display were followed by the amplification and transformation of EBY100 yeast cells with the DNA sequences corresponding to binding monobodies. The yeast cells were next sorted using FACS based on a strict gating strategy comprising double positive cells for monobody display and binding to biotinylated target. Isolated clones were next sequenced and cloned into the pHFT2 vector, digested with BamHI and XhoI, by a standard Gibson assembly protocol (NEB) for further characterization. Here, the monobody fragments were amplified from the isolated yeast plasmids via PCR using 5′-gtgaaaacctgtacttccagggatccatggctgcttcttctg-3′

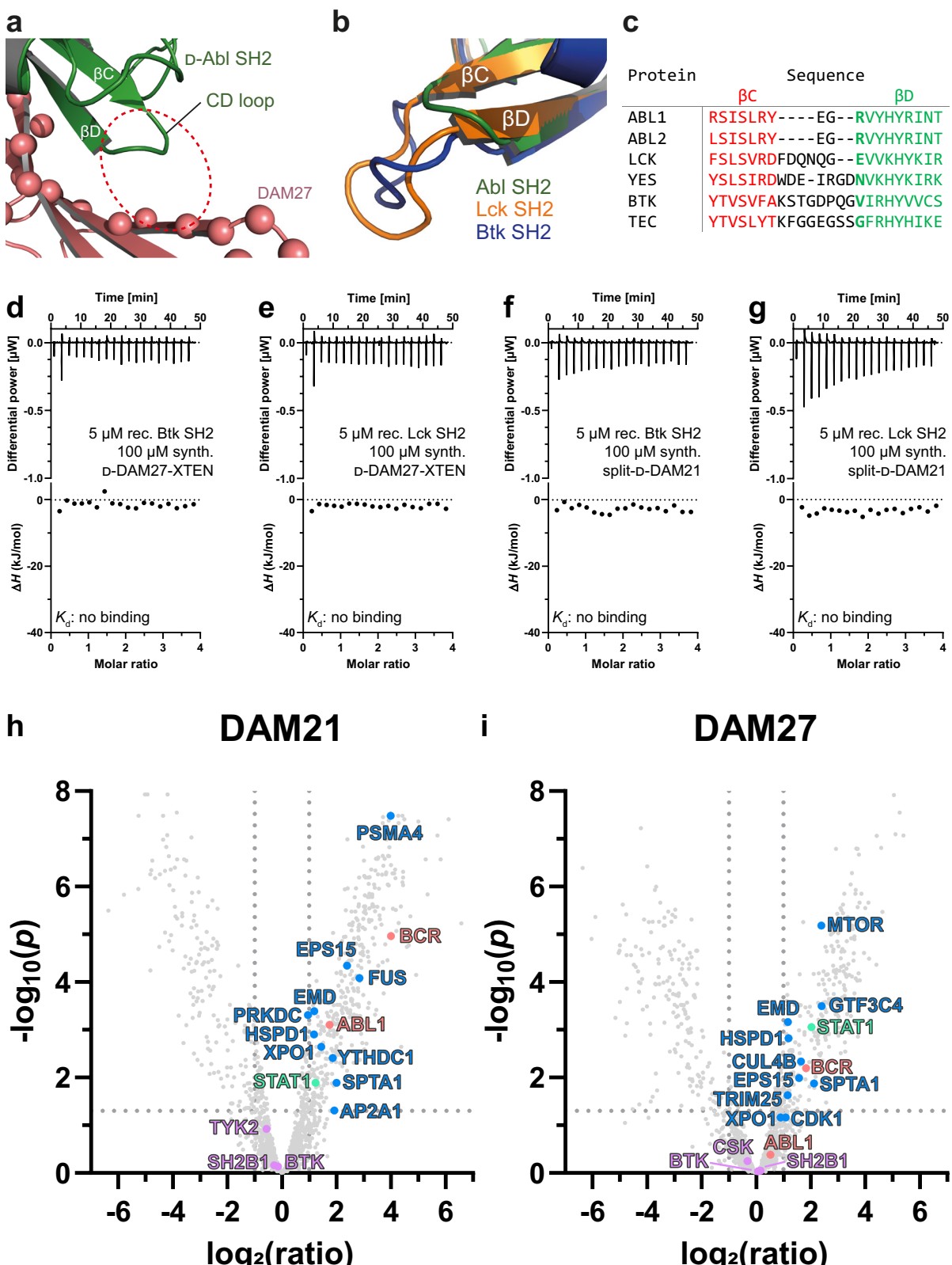

and 5′- cagtggtggtggtggtggtgctcgagctaggtacggtagttaatc-3′ as primers. A detailed description of the selection procedure can be found in the Supplementary Information.

**Binding assay in yeast display format**

Increasing concentrations of synthetic biotinylated protein target D-Abl SH2 resuspended in Tris-buffered saline (TBS) with 0.1% BSA

were incubated with $5 \times 10^5$ yeast cells per sample displaying monobodies and a mouse anti-V5 antibody (5 μL of 1:300 dilution) in a total volume of 20 μL for 30 min at room temperature. Three washes with TBS with 0.1% BSA followed together with incubation for 30 min with a streptavidin-DyLight650 and a FITC-coupled anti-mouse IgG (both diluted 1:1000 in the same tube and 20 μL used per sample) at room temperature in the dark. Samples were washed twice with

**Fig. 9 | Selectivity of D-DAM21/27 for the BCR::ABL1 SH2 domain. a** L-DAM27: D-Abl SH2 complex structure (PDB code: 9F00) highlighting the short Abl SH2 CD loop that enables binding to DAM27. The space where longer CD loops in other SH2 domains would be located is indicated by the dotted red circle. **b** Structural superposition of the Abl, Lck and Btk SH2 domains (PDB entries 3K2M, 1LKK and 6HTF, respectively). The additional four to six amino acid residues in the Lck and Btk SH2 CD loops are not compatible with binding to the DAM21 and DAM27 monobodies. **c** Multiple sequence alignment of the βC and βD strands of SH2 domains belonging to different SH2 domain-containing tyrosine kinase families. The βC and βD strands are colored in red and green, respectively, while the CD loop in between is black. Arg189 of ABL1, which is important for D-monobody binding via ionic interactions, is not conserved and colored in bold green to highlight the different amino acid sequences in this position. **d, e** Isothermal titration calorimetry (ITC) measurements of D-DAM27-XTEN titrated to the SH2 domains of (**d**) Btk and (**e**) Lck. **f, g** ITC measurements of split-D-DAM21 titrated to the SH2 domains of (**f**) Btk and (**g**) Lck. Each panel shows the raw heat signal of an ITC experiment (top)

and the integrated calorimetric data of the area of each peak (bottom). **h, i** Volcano plots of identified proteins via mass spectrometry after pulldown from K562 cell lysates comparing synthetic (**h**) split-D-DAM21 with split-L-DAM21 and (**i**) D-DAM27-XTEN with L-DAM27-XTEN. Pulldowns were performed in three biological replicates for each monobody. Protein identification and statistical analysis of replicates were done with MaxQuant 2.5.1.0 and Autonomics (R package version 1.13.21)[84] resulting in FDR-corrected $p$-values (Benjamini-Hochberg procedure) for each identified protein where a $p$-value below 0.05, which corresponds to $-\log_{10}(p)$ above 1.3, was considered statistically significant (dotted line intersecting $y$-axis). A $\log_2$(ratio) above 1.00 of D- vs. L-monobody correlates to a ratio above 2.00 and a higher abundance of the protein when the pulldown was performed with the D-monobody compared to its L-counterpart (dotted lines intersecting $x$-axis). The color coding represents the protein groups BCR and ABL1 (light red), BCR::ABL1/ABL1 interactors (blue), BCR::ABL1/ABL1 interactors with SH2 domains (green) and other proteins containing SH2 domains (purple). Plotted values of all highlighted protein groups are listed in Tables S9, S10.

Tris-buffered saline (TBS) with 0.1% BSA and analyzed on a Guava easyCyte flow cytometer (Luminex). Data were fitted on a 1:1 binding model using the Prism software (GraphPad) to determine $K_d$ values.

### Recombinant protein expression

Generally, *E. coli* BL21 Star (DE3) cells were transformed with plasmids containing the desired proteins. An overnight pre-culture was added to medium (1:20, $v/v$) and cells were grown at 37 °C, 200 rpm. For expressions in lysogeny broth (LB; until $OD_{600}$ 0.5–0.8) and terrific broth (TB; until $OD_{600}$ 1.2) medium, the culture was next induced with 0.5 mM IPTG final concentration and cells were further grown overnight at 18 °C, 200 rpm. Expressions in auto induction (AI) medium were placed at 18 °C without IPTG addition. The next day, cells were harvested at 6000×$g$ for 10 min at 4 °C.

The Abl SH2, Btk SH2 and Lck SH2 domains were produced in LB medium with an N-terminal tag including a His$_6$ and Glutathione S-transferase (GST), and tobacco etch virus (TEV) protease recognition motifs using the pETM30 vector. The Abl SH2 A198C mutation was introduced using the QuikChange II Site-Directed Mutagenesis Kit (Agilent) with 5′-cattacaggatcaacacttgctctgatggcaagctc-3′ and 5′-gagcttgccatcagagcaagtgttgatcctgtaatg-3′ as primers. Protein purification and tag cleavage was performed as described previously by nickel-affinity chromatography (His-Trap FF crude, Cytiva) and SEC (HiLoad 16/600 Superdex 75 pg, Cytiva) on an Äkta Avant or Äkta go system (Cytiva)[34].

The Abl KD and SH2-KD proteins in pET21d vectors containing a C-terminal His$_6$ tag were each co-expressed with YopH phosphatase in TB medium and purification was performed as described previously using nickel-affinity chromatography (His-Trap FF crude, Cytiva) and anion exchange chromatography (Mono Q 5/50 GL, Cytiva) on an Äkta Avant system (Cytiva)[13,52].

All recombinantly expressed monobodies were produced with an N-terminal tag containing His$_{10}$, FLAG and a TEV recognition motif. Purification was carried out on Ni-NTA Agarose beads (Qiagen) with 1 mL bead volume by gravity flow and subsequent SEC purification (HiLoad 16/600 Superdex 75 pg, Cytiva) into 20 mM HEPES, 150 mM NaCl, 0.5 mM TCEP-HCl on an Äkta Avant or Äkta go system (Cytiva). The D83A/E85A mutation of DAM21 was introduced using the QuikChange II Site-Directed Mutagenesis Kit (Agilent) with 5′-ctgactactacgcggggtgcgatctcttggtac-3′ and 5′-gtaccaagagatcgcacccgcgtagtagtcag-3′ as primers. The XTEN peptide extension of DAM27 was introduced through standard Gibson assembly cloning (NEB) using a DAM27 fragment containing the XTEN peptide ordered from Twist Bioscience.

### Biotinylation of recombinantly expressed proteins

Recombinantly expressed monobodies were biotinylated using two approaches: Either the monobody contained an Avi tag

(-GLNDIFEAQKIEWHE-) and was biotinylated with BirA in a reaction mixture containing 952 μL 100 μM monobody in PBS, pH 7.4, 5 μL 1 M MgCl$_2$, 20 μL 100 mM ATP, 20 μL 50 μM BirA, 3 μL 50 mM D-biotin. The reaction proceeded first for 3 h at 30 °C, then fresh BirA and D-biotin were added and incubated for 3 h at 30 °C. The reaction mixture was purified via SEC on a Superdex 75 Increase 10/300 GL column. Another biotinylation approach included coupling of biotin-maleimide to a cysteine residue of the monobody. Here, the reaction mixture contained 1 mL 70 μM monobody in PBS, pH 7.4 and 7 μL 100 mM biotin-maleimide (B1267-25MG, Sigma-Aldrich) in DMSO. After 2 h at rt, excess biotin-maleimide was removed and the storage buffer was exchanged using 5 mL Pierce Dye & Biotin Removal Spin Columns (Thermo Fisher Scientific) according to the manufacturer's instructions.

### Competitive fluorescence polarization (FP) assay

A pY-peptide known to bind SH2 domains of Src family kinases with the sequence 5-carboxyfluoresceine-EPQpYEEIPIYLK-CONH$_2$ ordered from Peptide Synthetics (UK) was used in D-configuration. A 100 μL solution consisting of the peptide (50 nM) mixed with synthetic D-Abl SH2 (10 μM) followed by an addition of recombinantly expressed L-monobodies DAM21, DAM26, DAM27, DAM28, DAM30.1 and DAM30.2 (20 μM) in 20 mM HEPES, 150 mM NaCl, 0.5 mM TCEP-HCl and 0.02% Triton-X, pH 7.4. For comparison, binding reactions were also set up for recombinantly expressed L-Abl SH2 (10 μM) complexed with the L-pY peptide (50 nM) and addition of recombinantly expressed L-monobodies AS25, HA4 and HA4 Y87A (20 μM) in the same buffer. Measurements were performed in FP compatible dark bottom 96-well plates (675076, Greiner Bio-One) on a M5 plate reader from Molecular Devices at room temperature. FP data was acquired using 492 nm as excitation and 525 nm as emission wavelength with a filter at 515 nm. Raw data were subtracted from measurements of peptide complexed to Abl SH2 without monobodies in solution.

### Isothermal titration calorimetry (ITC)

Recombinant and synthetic proteins were used in ITC buffer (20 mM HEPES, pH 7.4, 150 mM NaCl, 0.5 mM TCEP-HCl). Protein concentration was measured on a NanoDrop 2000c (Thermo Scientific). ITC measurements were acquired on a MicroCal PEAQ-ITC instrument (Malvern Panalytical) and consisted of the titration of the monobody solution from the syringe in 19 steps with 0.4 μL for the first injection followed by 2.0 μL for the remaining steps with a spacing of 150 s between injections at 25 °C. The reference power, feedback and stir speed were set to 3.00 μcal/s, high and 750 rpm, respectively. Protein concentrations were used as indicated in each figure. Thermodynamic parameters were determined with the MicroCal PEAQ-ITC analysis software.

## X-ray crystallography

Complex formation was performed by mixing TEV-cleaved DAM21 or DAM27 (1.5 eq.) with refolded, synthetic D-Abl SH2 A198C (1 eq.) and subsequent purification of each complex by SEC on a HiLoad 16/600 Superdex 75 pg column (Cytiva) column using the Äkta Avant system (Cytiva) into 20 mM HEPES, 150 mM NaCl, 0.5 mM TCEP·HCl. The complexes were concentrated in Amicon Ultra-15 centrifuge filter tubes (MWCO: 3.5 kDa) to 7.3 mg/mL (with DAM21) and 6.9 mg/mL (with DAM27) before crystallization trials were performed in sitting drop plates by mixing 100 nL of protein with 100 nL of buffer conditions. The best diffracting crystals were obtained in 0.16 M ammonium sulfate, 0.08 M sodium acetate pH 4.6, 20% (w/v) PEG 4000, 20% (v/v) glycerol (DAM21) and 0.8 M tri-sodium citrate pH 6.5, 20% (v/v) ethylene glycol (DAM27) and frozen in 15% (v/v) glycerol in liquid nitrogen.

X-ray data were collected at 100 K using the beamlines X06SA and ID23-2 of the Swiss Light Source (SLS), Villigen, Switzerland, and the European Synchrotron Radiation Facility (ESRF), Grenoble, France, respectively. After data processing by XDS[79], the structures were determined by molecular replacement using PHASER[80], the inverted structure of the Arg SH2 domain (PDB code: 4EIH) and AlphaFold models of the monobodies as initial search models. Manual model rebuilding was done with Coot[81] prior to subsequent refinement with phenix.refine[82]. Overall data and refinement statistics are summarized in Table 1.

## Solid-phase peptide synthesis (SPPS), native chemical ligation (NCL) and desulfurization

Peptides were synthesized by standard Fmoc-strategy on a ResPep SLi (Intavis) parallel synthesizer using Fmoc- and sidechain-protected amino acids in either L- or D-configuration obtained from Carbolution. Native chemical ligations were generally performed in 6 M guanidine-HCl, 200 mM Na$_2$HPO$_4$, 100 mM MPAA and 20 mM TCEP·HCl, pH 7.0 where the N-terminal peptide with MeNbz-linker was used at 2 mM (1 eq.) and the C-terminal peptide at 1.2 eq for at least 24 h. Desulfurizations of synthetic monobodies were carried out in 6 M guanidine-HCl, 200 mM Na$_2$HPO$_4$, 375 mM TCEP·HCl, 150 mM MESNa, 115 eq. VA-044, pH 7.5 with the peptide at 325 μM for 16 h. Detailed synthesis protocols can be found in the Supplementary Information.

## Refolding of synthetic monobodies

The polypeptides were dissolved in solubilization buffer (6 M guanidine-HCl, 500 mM arginine-HCl, 20 mM HEPES, pH 8.5, 150 mM NaCl, 0.5 mM TCEP·HCl) at a concentration of 0.5 mg/mL and transferred into a Slide-A-Lyzer G2 dialysis cassette with a cut-off of 3.5 kDa (Thermo Scientific). The solution was dialyzed against a 200-fold volume of refolding buffer 1 (500 mM arginine-HCl, 20 mM HEPES, pH 7.4, 150 mM NaCl, 0.5 mM TCEP·HCl) for 2 h at 4 °C. Then, the buffer was replaced twice for refolding buffer 2 (20 mM HEPES, pH 7.4, 150 mM NaCl, 0.5 mM TCEP·HCl) and the dialysis was carried out for another 2 and 16 h. Afterwards, the solution containing the refolded protein was purified via SEC in refolding buffer 2 on the Äkta Avant or Go system (Cytiva) with a Superdex 75 Increase 10/300 GL or HiLoad 16/600 Superdex 75 pg column (Cytiva) to check for and remove aggregates at a flow rate of 0.5 or 1 mL/min. The fractions containing the synthetic refolded and monomeric proteins were collected, concentrated, flash-frozen in liquid nitrogen and stored at −80 °C.

## Circular dichroism (CD) spectroscopy

The recombinant and synthetic proteins originally in HEPES buffer (20 mM HEPES, pH 7.4, 150 mM NaCl, 0.5 mM TCEP·HCl) were dialyzed three times against a 200-fold volume of phosphate-buffered saline (PBS) at pH 7.4, twice for 2 h and then for 16 h at 4 °C. Dialysis was carried out in Slide-A-Lyzer G2 dialysis cassettes (cut-off 3.5 kDa, Thermo Scientific). CD spectra were recorded in a quartz cuvette (path length: 0.1 cm, Hellma Analytics) containing 20 μg of sample in 300 μL

PBS buffer (pH 7.4) on a JASCO J-815 circular dichroism spectrometer at 20 °C and a data interval of 0.1 nm. The mean residue ellipticity (MRE) was calculated according to T. E. Creighton[83], secondary structure predictions were calculated with BeStSel[51] and the data was plotted using the software GraphPad Prism 8.

## Nano differential scanning fluorimetry (nanoDSF)

The thermal denaturation curves were determined by measurements of intrinsic tryptophan fluorescence. This analysis was performed using label-free, native differential scanning fluorimetry on a Prometheus NT.48 instrument (NanoTemper). Approximately 10 μL of the

**Table 1 | Data collection and refinement statistics**

| | DAM21·D-SH2 complex (9FO1) | DAM27·D-SH2 complex (9FO0) |
|---|---|---|
| Source | SLS, X06SA | ESRF, ID23-2 |
| Wavelength (Å) | 1.00003 | 0.8731 |
| Resolution range | 48.15–2.73 (2.81–2.73) | 48.96–2.91 (3.01–2.91) |
| Space group | $P\,2_1$ | $P\,6_5\,2\,2$ |
| Unit cell (Å,°) | 76.64 62.29 116.66 90 98.06 90 | 111.33 111.33 205.75 90 90 120 |
| Total reflections | 93,888 (4103) | 677,041 (67967) |
| Unique reflections | 25,843 (1293) | 17,328 (1672) |
| Multiplicity | 3.6 (3.2) | 39.1 (39.9) |
| Completeness, spherical (%)[a] | 87.9 (51.1) | 99.7 (98.1) |
| Completeness, ellipsoidal (%)[a] | 95.7 (88.7) | n.a. |
| Mean I/sigma(I) | 6.9 (2.3) | 15.7 (0.8) |
| Wilson B-factor (Å$^2$) | 36.76 | 84.31 |
| R-merge | 0.146 (0.810) | 0.270 (3.634) |
| R-meas | 0.171 (0.968) | 0.273 (3.680) |
| R-pim | 0.087 (0.520) | 0.043 (0.577) |
| CC1/2 | 0.987 (0.530) | 0.999 (0.526) |
| Refl. used in refinement | 25,777 (3412) | 17,285 (1672) |
| Refl. used for R-free | 644 (39) | 859 (82) |
| R-work | 0.2221 (0.2370) | 0.2094 (0.3647) |
| R-free | 0.2704 (0.2981) | 0.2565 (0.3794) |
| Non-hydrogen atoms | 6102 | 3012 |
| Macromolecules | 5972 | 2999 |
| Ligands | 4 | 0 |
| Solvent | 126 | 13 |
| Protein residues | 768 | 387 |
| RMSD bonds (Å) | 0.007 | 0.010 |
| RMSD angles (°) | 1.17 | 1.34 |
| Ramachandran favored (%) | 96.98 | 94.59 |
| Ramachandran allowed (%) | 2.75 | 2.70 |
| Ramachandran outliers (%) | 0.27 | 2.70 |
| Rotamer outliers (%) | 2.81 | 6.41 |
| Clashscore | 10.34 | 14.13 |
| Average B-factor (Å$^2$) | 39.65 | 87.73 |
| Macromolecules (Å$^2$) | 41.69 | 89.20 |
| Ligands (Å$^2$) | 38.20 | – |
| Solvent (Å$^2$) | 27.11 | 68.01 |
| TLS groups | 8 | 4 |

Statistics for the highest-resolution shell are shown in parentheses.

[a]DAM21·D-SH2 dataset was corrected by STARANISO[85] for anisotropic diffraction. Cut-offs used direction 0.477 a* + 0.511 b* + 0.715 c* for best diffraction (2.73 Å), −0.725 a* + 0.689 c* for worst diffraction (3.44 Å).

protein samples at a concentration of 0.07 mg/mL in PBS (pH 7.4) were loaded in Prometheus NT.48 capillaries. The tryptophan residues of the proteins were excited at 280 nm, and the fluorescence intensity was recorded at 330 and 350 nm. Excitation power was set to 50%, and the temperature of the measurement compartment increased from 20 to 95 °C at a rate of 1 °C/min. Melting temperatures ($T_m$) were determined by the Prometheus software through calculation of the fluorescence ratio at 330 and 350 nm and of the first derivative. The data was plotted using the software GraphPad Prism 8.

### Protease digest assay
Proteins were used at 0.5 mg/mL and incubated with 0.025 mg/mL of proteases Pepsin (dissolved in Milli Q water; 1071920001, Sigma-Aldrich) or Proteinase K (dissolved in 50 mM Tris-HCl, pH 8.0, 1 mM CaCl$_2$; P2308-25MG, Sigma-Aldrich) (protein-to-protease ratio of 20:1) in 40 µL per reaction. Digests with Pepsin were performed in 20 mM HEPES, 150 mM NaCl, 0.5 mM TCEP-HCl at a pH of 2.0 and 37 °C. Digests with Proteinase K were performed in 50 mM Tris-HCl, pH 8.0, 1 mM CaCl$_2$ at 37 °C. After 0, 4 and 24 h, 10 µL samples were taken for SDS-PAGE analysis.

### Plasma stability assay
Monobody stocks were prepared at a concentration of 50 µM in PBS, pH 7.4. 60 µL of stocks were subsequently mixed at a volume ratio of 1:1 with either mouse plasma (plasma) or PBS (control) and incubated at 37 °C. At indicated time points, 20 µL samples were diluted 1:5 in PBS. 20 µL of this dilution were mixed with 10 µL of 4X Laemmli SDS buffer (400 mM DTT, 8% SDS, 200 mM Tris-HCl, pH 6.8, 40% Glycerol, 0.02% bromophenol blue) and denatured at 95 °C for 5 min. 25 µL of these samples were separated using SDS-PAGE and blotted onto a 0.2 µm Amersham Protran nitrocellulose membrane (10600004, Cytiva). Monobodies were detected using IRDye680-coupled Streptavidin (1:10,000 dilution, 926-68079, LiCOR).

### Radiometric kinase assay
Recombinantly expressed proteins Abl KD or SH2-KD, each at 50 ng, were preincubated with the recombinant or refolded, synthetic monobodies at 5 µM for 10 min at room temperature in a volume of 10 µL. Kinase activity was determined by the addition of 25 µM ATP, 3 µCi of [γ-$^{32}$P]-ATP (SRP-301, Hartmann Analytic), and 100 µM of an optimal Abl substrate sequence carrying an N-terminal Biotin (Biotin-GGEAIYAAPFKK-amide) in kinase assay buffer (20 mM Tris-HCl, pH 7.5, 10 mM MgCl$_2$,1 mM DTT) for 15 min at room temperature in a final assay volume of 20 µL. 8 µL of each reaction terminated with 10 µL of 7.5 M Guanidine-HCl were spotted onto a SAM2 Biotin Capture membrane square (V2861, Promega) and further treated according to the instructions of the manufacturer. $^{32}$P radiation of membrane squares was measured on the scintillation counter Hidex 300 SL by counting Cerenkov radiation for 1 min per sample. Data was analyzed using GraphPad Prism 8.

### Flow cytometry analysis of monobodies binding to Bcr-Abl in K562 cells
K562 suspension cells were cultivated in RPMI medium supplemented with 10% FBS and Penicillin/Streptomycin and split every 2–3 days. For flow cytometry analysis, cells were taken up in PBS, pH 7.4, at a density of $5 \times 10^5$ cells/mL and fixed in 3.2% Paraformaldehyde (PFA; $v/v$) by addition of 200 µL PFA solution (E15710, Science Services) for every 1 mL of cell suspension and incubated for 10 min at room temperature (rt). The suspension was centrifuged (5 min, 500×$g$), the pellet was slowly resuspended in ice-cold methanol at a density of $2.5 \times 10^5$ cells/mL while vortexing, placed for 20 min on ice and then stored at −20 °C overnight. Then, cells were washed three times with the same amount of PBS, resuspended in FACS buffer (4% FBS in PBS, pH 7.4) at a density of $1 \times 10^6$ cells/mL and placed for 2 h on ice. Afterwards, the cell

suspension was split and $5 \times 10^5$ cells were used per sample, which were incubated with 20 µL FcBlock (diluted 1:20 ($v/v$) in FACS buffer; BUF070, Bio-Rad) for 10 min at rt. Then, monobody solutions were added at a concentration of 2 µM in 50 µL total volume and incubated for 45 min at rt. 1 mL FACS buffer was added to each sample, the samples were centrifuged (5 min, 500×$g$) and the supernatant was removed. 5 µL of AlexaFluor488-coupled Streptavidin (diluted 1:1000; S32354, Thermo Fisher Scientific) in 45 µL FACS buffer was added to each sample, incubated for 45 min at rt in the dark, then mixed with 2 mL FACS buffer, centrifuged (5 min, 500×$g$), taken up in 200 µL PBS and placed on ice until measurement. Fluorescence intensity of samples was analyzed on a Guava easyCyte flow cytometer (Luminex). Data was analyzed using GraphPad Prism 8.

### K562 cell lysis, BCR::ABL1 pulldown and mass spectrometry
K562 cells were harvested by centrifugation (500×$g$, 5 min) at a density of ~1.0 × 10$^6$ cells/mL. The cell pellet was washed with PBS, frozen in liquid nitrogen and stored at −80 °C until lysis. The cells were lysed for 5 min on ice in IP buffer (50 mM Tris-HCl, pH 7.5, 150 mM NaCl, 1% NP-40, 5 mM EDTA, 5 mM EGTA) containing additionally 50 mM NaF, 1 mM vanadate, 1 mM PMSF, 10 µg/mL TPCK and 1x protease inhibitor cocktail (Roche) and centrifuged for 10 min at 20,000×$g$ and 4 °C. The total protein amount of the supernatant was determined by standard Bradford assay. The pulldown was performed in three biological replicates for each monobody on Streptavidin MagneSphere Paramagnetic Particles (Z5481, Promega) where 200 µL of beads were used per sample. The beads were washed twice with IP buffer, then 1.5 µg of each monobody were added to the beads and incubated for 2 h at 4 °C. After washing the beads once with IP buffer, the K562 cell lysate containing 3 mg of total protein was added to each sample and incubated for 16 h at 4 °C. The beads were washed three times with IP buffer without NP-40 (50 mM Tris-HCl, pH 7.5, 150 mM NaCl, 5 mM EDTA, 5 mM EGTA). Trypsin (0.1 µg in 50 µL 50 mM ammonium-bicarbonate buffer, pH 8.0) was added to the beads and samples were incubated at 37 °C for 45 min. Subsequently, the supernatant was transferred into fresh tubes and digested overnight at 37 °C to completeness. For the reduction of disulfide bridges 5 mM DTT was added. Samples were then incubated for 15 min at 95 °C. Subsequently, the resulting sulfhydryl groups were chemically modified by adding iodoacetamide to a final concentration of 25 mM and incubating samples for 45 min at RT in the dark. Excess iodoacetamide was quenched by the addition of 50 mM DTT and incubation for one more hour at RT. Reduced and alkylated peptides were then desalted and concentrated using Chromabond C18WP spin columns (730522, Macherey-Nagel) according to manufacturer protocols. Finally, peptides were dissolved in 20 µL of water with 5% acetonitrile and 0.1% formic acid. The mass spectrometric analysis of the samples was performed using a timsTOF Pro mass spectrometer (Bruker Daltonic). A nanoElute HPLC system (Bruker Daltonics), equipped with an Aurora C18 RP column (25 cm × 75 µm ID) filled with 1.7 µm beads (IonOpticks, Australia) was connected online to the mass spectrometer. A portion of 2 µL of the peptide solution was injected directly on the separation column. Sample loading was performed at a constant pressure of 800 bar. Separation of the tryptic peptides was achieved at 60 °C column temperature with the following gradient of water/0.1% formic acid (solvent A) and acetonitrile/0.1% formic acid (solvent B) at a flow rate of 400 nL/min: Linear increase from 2% B to 17% B within 18 min, followed by a linear gradient to 25% B within 9 min and linear increase to 37% solvent B in additional 3 min. Finally, B was increased to 95% within 10 min and held at 95% for additional 10 min. The built-in "DDA PASEF-standard_1.1sec_cycletime" method developed by Bruker Daltonics was used for mass spectrometric measurement. Data analysis was performed using MaxQuant 2.5.1.0 (MPI of Biochemistry, Germany) and statistical analysis of replicates was done with Autonomics (R package version 1.13.21)[84].

**Reporting summary**

Further information on research design is available in the Nature Portfolio Reporting Summary linked to this article.

## Data availability

The crystal structures of the DAM27- and DAM21-ᴅ-Abl SH2 complexes were deposited at Protein Data Bank 9F00 and 9F01. The mass spectrometry proteomics data have been deposited to the ProteomeXchange Consortium via the PRIDE partner repository with the dataset identifier PXD056009. Supplementary Information is provided with this paper. Source data are provided with this paper.

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

## Acknowledgements

We thank all members of the Vázquez and Hantschel labs for input and discussions, O. Stehling for the use of the CD spectrometer and nanoDSF device, R. Pöschke and T. U. Hedderich of MarXtal for assistance in protein crystallization. We acknowledge support by the European Research Council (Consolidator Grant; ERC-2016-CoG 682311) to O.H., N.S., A.Ku. and A.V.D.-F., and the European Synchrotron Radiation Facility (ESRF) and

Swiss Light Source (SLS) for provision of synchrotron radiation facilities; we like to thank L. McGregor and T. Tomizaki for assistance and support in using beamlines ID23-1 and X06SA, respectively.

## Author contributions

N.S. planned, conducted and analyzed most experiments. A.Ku. performed monobody selection and contributed to data analysis, L.K. performed protein crystallography, A.V.D.-F. performed plasma stability experiments, F.A. provided vital tools and expertise on peptide synthesis design, U.L. processed proteomics samples and performed mass spectrometry analysis, A.Ko. and M.R.-B. provided training and troubleshooted monobody selection. S.K. contributed to the study design and data interpretation. L.-O.E. performed crystallography data analysis and solved the crystal structures. O.V. and O.H. designed and coordinated the study, planned the experiments and analyzed data. O.H. and N.S. wrote the manuscript. All authors edited the manuscript.

## Funding

## Competing interests

A.Ko. and S.K. are listed as inventors on issued and pending patents on the monobody technology filed by the University of Chicago (US Patent 9512199 B2 and related pending applications). The other authors declare no competing interests.
