## [Transparent Peer Review file · Nature Communications]

Development of mirror-image monobodies targeting the oncogenic BCR::ABL1 kinase

Corresponding Author: Professor Oliver Hantschel

Version 0:

Reviewer comments:

Reviewer #1

(Remarks to the Author)

"Development of mirror-image monobodies targeting the oncogenic BCR::ABL1 kinase" by Schmidt et al reports on the development of D-monobodies that bind the SH2 domain of BCR::ABL1. The core idea of the paper – using a chemically synthesized D-protein receptor as a tool to discover D-peptide ligands of the naturally occurring L-protein receptor – was first reported almost 30 years ago. Nevertheless, the authors of the current manuscript push this venerable idea to novel technical limits for the discovery of D-proteins (as opposed to just peptides) as potential therapeutic agents.

This process involves overcoming an array of technical challenges: 1.) chemical synthesis of the D-version of the target protein (in this case, the SH2 domain of BCR::ABL1); 2. selection of L-proteins (in this case, monobodies) with high affinity for the D-target, 3.) chemical synthesis of the D-monobodies, which, by well established principles of chirality, will have equal affinities to the natural L-target that the corresponding L-monobodies have to the D-target.

The potential payoff of this labor-intensive approach is that D-monobodies would be predicted to have heightened plasma stability and resistance to degradation by proteases, as compared to L-monobodies, potentially furnishing protein therapeutics with superior pharmacokinetic properties.

Carrying out a prodigious amount of peptide synthesis, the authors show quite convincingly that the approach can work. They produce two D-monobodies (D-DAM21 and D-DAM27) and demonstrate their binding to the BCR::ABL1 SH2 domain. They also solve the crystal structures of the corresponding L-monobodies bound to the D-SH2. Somewhat frustratingly, the crystal structures show two different monobody/SH2 interfaces, yet the authors make a convincing case that they are able to determine which interface (interface 2) is the one relevant to the complex in solution.

On the whole, the biochemical experiments in the paper are clear and convincing. However, the manuscript contains several limitations:

1. Lack of monobody cell permeability: It is puzzling that the authors chose an intracellular target (SH2 domain of BCR::ABL1) as their target for their D-monobody approach, when there is no reason to expect that the monobodies would be cell permeable. Nothing can be done about this at this point, but the authors should discuss their target choice a bit more. Why did they choose the BCR::ABL1 SH2 when an extracellular protein domain would be the more intuitive choice for a D-monobody target?
2. Lack of demonstrated selectivity by D-monobodies: The current manuscript contains no experiments that establish the selectivity for the BCR::ABL1 SH2 domain over other SH2 domains (or other non-SH2 protein domains, for that matter). The authors assume selectivity based on their selection, which utilized the SH2 from BCR::ABL1, but they should test this directly. Do D-DAM21 and D-DAM27 selectively bind BCR::ABL1 SH2 over other SH2 domains? If so, how selective are they? Of course, the authors need not measure binding constants for all other SH2 domains, but a few judiciously chosen SH2s (maybe the three most homologous?) would do the trick.
3. Uninformative experiments in permeabilized cells: As noted before, the authors' D-monobodies are not cell permeable, so the authors must resort to using permeabilized cells to test whether the D-monobodies can engage the target SH2 in a complex mixture of proteins (Figure 9C). Unfortunately, this experiment does not establish target engagement, as the fluorescence readout of the assay does not distinguish between on-target and off-target binding. In fact, off-target binding in this experiment would give higher signal, so a "big bar" does not provide evidence of binding to BCR::ABL1 SH2. Largely because this experiment does not distinguish between on-target and off-target binding, the data raise more questions than

they answer: Why does the previously established Abl binder HA4 not give rise to a signal increase? Why do the L-DAMs (which shouldn't bind BCR::ABL1 SH2) give rise to signal increases? Why are the signal increases of the D-DAMS so much larger than those from the previously established BCR::ABL1 binders? (This reviewer's guess is that the huge increases for the D-DAMs is due to off-target binding.)

In short, this experiment is uninformative. The figure should be removed, and the authors should figure out a better way to probe target engagement in a complex proteome. Maybe a better experiment would be to incubate immobilized D-monobodies with a crude lysate and quantify the proteins that bind by mass spec. In addition to establishing that the D-monobodies can bind their target in a complex proteome, this would help address the selectivity question raised above.

Reviewer #2

(Remarks to the Author)

We all know about the possibility in life to create a mirror-image of proteins and ligands. While D-enzymes would recognise and modify D-target partners, the recognition of the L-target would become problematic. In this work Nina Schmidt et al. demonstrated the possibility to identify affinity reagents in the L-form that associate to a synthesised D-target protein with high affinity (11-70 nM). Thereafter as expected, they show that the D-affinity reagent also recognises the native L-target protein. This is important as the D-affinity reagent is expected to be stable in plasma and non-immunogenic, so as to become an improved therapeutic agent.

The use of a synthetic library of monodies (stable, small, monomeric domains lacking disulfide bonds, ability to refold from linear polypeptide) seems to be a clever choice, although DARPinS or affibodies could have been valid alternatives.

The crystal structure of two L-monobodies in complex to the D-Abl-SH2 target has been solved and shows that the binding is overlapping with a D-pY peptide binding site.

The work has been explained in a logical and clear manner. Although there were apparently problems to synthesize the D-alternative of the affinity reagent, the authors found an elegant solution by generating a split variant.

In view of the variable and abundant amount of work shown, my critics to the work are minor:

- The last part of the last sentence of the abstract "... enabling their use as future D-protein therapeutics to target a broad spectrum of protein-protein interactions" is not supported by the current work:
First, the split monobody is probably too unstable to be used as a therapeutic,
Secondly, more importantly, to have a biological effect they use a concentration of 5 microM added to permeabilised cells. This is still far from a practical solution. Nowhere was there an indication of how to get the synthesised D-protein inside cells. Perhaps it might be a possibility if the target recognition would occur on the cell membrane, rather than cytoplasmic.

It was also confusing to note that the "split L-DAM21 showed inhibition as compared to its L-counterpart" (Line 352-353). Does this mean that the L-DAM21 binds to D-Abl-SH2 as well as to L-Abl-SH2??? or does it mean that neither of them bind sufficiently to have an impact on enzyme inhibition (in contrast to D-DAM27). The binding study (Fig9C) suggests (to me) that indeed the L DAM21 doesn't bind significantly to its D target, in contrast to the binding of the split D-DAM21. Could it be that the split D-DAM21 is binding to the L target, but the D-DAM21 is too unstable to exert a kinase inhibition.

Reviewer #3

(Remarks to the Author)

This manuscript describes the identification and characterization of D-monobodies against the BCR-ABL1 SH2 domain. As a very small antibody-mimicking domain, monobodies are particularly well-suited for synthesis in mirror-image. The first step is synthesis of a the mirror-image BCR-ABL1 SH2 domain (following their previous L-synthesis). Monobody libraries are then screened using a combination of initial phage display with polishing rounds via yeast display. 6 clones were identified and the two highest affinity clones were further characterized (by crystallography, ITC, competition FP). These two monobodies were synthesized in D (one by NCL & the other as a pair of complementing fragments) and further characterized, including proteolytic resistance and inhibition of kinase activity (in vitro and in permeabilized cells). Overall, this is an interesting, thorough, and clearly presented paper that advances the case for D-monobodies as a promising emerging class of inhibitors. However, there are some additional controls that would strengthen the conclusions and some minor technical issues that should be addressed (listed below).

-it would be helpful to add a brief discussion about how a D-monobody will access an intracellular target, especially in the context of a mirror-image monobody (fully synthetic)
-no details on the progression of library selection were provided - a brief summary in the supplement would be helpful (e.g., progress of rounds, sequencing details).
-the case for specificity would be strengthened by adding controls to measure specificity towards the BCR-ABL1 SH2 domain (vs. other SH2 domains)
-L-DAM21 & L-DAM27 bind surprisingly well (~50% of signal observed with D-DAM21/27) to BCR-ABL1 in permeabilized cells (despite showing minimal binding by BLI). It is unclear why this non-specific binding is so high - would be helpful if the authors could explain (or at least speculate).
-The D83A/E85A mutation control would have more value if the folding of this protein was confirmed (e.g., by CD).
-line 736: I think "rippled" was meant instead of "scribbled"

-FP curves (for Fig. 2F) should be shown in SI
-line 152: "elution volumes of all monobodies corresponded to their calculated monomeric molecular weights", but DAM28 elutes at a higher molecular weight
-In Fig. 6B, what is the source of the discrepancy between recombinant & synthetic DAM27-XTEN by SDS-PAGE?
-In Fig. 7, what is the source of the significant discrepancies in stoichiometry? Impurities in the synthetic constructs?
-In Fig. 8B, there are some irregularities on the gel: synth-L-DAM27-XTEN shows higher signal at 24 h than at 4 h & the D-DAM27-XTEN signal appears to be increased at 4 & 24 h (above input). Something similar is shown in Fig. 8C (for the D-protein).
-In Fig. 9B, the lack of inhibition seen with D-DAM21 is puzzling (inhibition compared to the slight stimulation seen with the L-DAM21 negative control is not convincing).

Version 1:

Reviewer comments:

Reviewer #1

(Remarks to the Author)

The authors were responsive to my comments on the previous manuscript, and the revised manuscript is significantly improved. In particular, the new experiments presented in Fig. 9D-G provide evidence for selectivity for the Abl SH2 domain over other SH2 domains, and I was pleased that the authors carried out the proteomic experiments that I suggested upon reviewing the original submission. Unfortunately, the proteomics results are disappointing and raise both technical questions and questions of interpretation that should be addressed in another round of revision.

Technical questions/issues:

- 1.) The authors state that the MS experiments were performed in triplicate ("Pulldown and mass spectrometry analysis were performed three times for each protein"), yet all the data are presented as single values without associated error bars. Are the presented data averaged between three experiments or taken from a single representative experiment? If averaged, the authors should include error bars and describe how the error values were calculated. If representative, the authors should state why they did not average and present the data from the other two experiments in Supplementary Information.
- 2.) How did the authors define and calculate "Significance" (y-axis in Fig. 9H-I)? Why is the significance identical for a protein between separate experiments? In other words, why are there single significance values for each protein (Table S9), as opposed to separate values for the DAM-21 and DAM27 experiments?
- 3.) The authors refer to the proteomic experiments as occurring "in cells" when the experiments were performed in cell lysates. This should be fixed prior to publication. (For example, "Binding of D-monobodies to Bcr-Abl in CML cells: In order to determine if D-monobodies can bind to full-length BCR::ABL1 in the context of the complex proteome of human cells, we performed...")

Question of interpretation:

The text in lines 397-420 does not really engage the proteomic data shown in Fig. 9H-I and Table S9, which reveal a disappointing lack of D/L selectivity. The data are actually quite troubling in regard to the manuscript's central hypothesis that the D-DAMs selectively bind to Abl in the context of a complex proteome. The data in Table S9 show that D-DAM27 and L-DAM27 pull down equal Abl in essentially equal amounts (D/L ratio = 1.09). Likewise, a few of the known Abl interactor proteins have D/L ratios of approximately 1 and some even have ratios of less than 1, indicating greater pull down with an L-monobody than the corresponding D-monobody (e.g., ABI, AB2M1, CBLB, CRKL). The authors should completely re-write the description of the proteomic experiments in a manner that accurately describes the selectivity complications revealed by the data.

Reviewer #2

(Remarks to the Author)

The authors took the comments and critics of the reviewers into consideration and replied in the best possible way.

Reviewer #3

(Remarks to the Author)

This revised manuscript has addressed the concerns raised in the original review (with extensive new supporting data, as well as additional caveats in the text). The addition of experimental specificity data is especially valuable. It remains a substantial body of work and, with the revisions, the conclusions are now robustly supported.

Version 2:

Reviewer comments:

Reviewer #1

(Remarks to the Author)

The authors have satisfactorily responded to my comments on the earlier version of the manuscript.

Point-by-point response to the reviewers' comments (NCOMMS-24-22129)

Our comments in red, **additional data/changes to manuscript/figures in bold red**

Referee #1

“Development of mirror-image monobodies targeting the oncogenic BCR::ABL1 kinase” by Schmidt et al reports on the development of D-monobodies that bind the SH2 domain of BCR::ABL1. The core idea of the paper – using a chemically synthesized D-protein receptor as a tool to discover D-peptide ligands of the naturally occurring L-protein receptor – was first reported almost 30 years ago. Nevertheless, the authors of the current manuscript push this venerable idea to novel technical limits for the discovery of D-proteins (as opposed to just peptides) as potential therapeutic agents.

This process involves overcoming an array of technical challenges: 1.) chemical synthesis of the D-version of the target protein (in this case, the SH2 domain of BCR::ABL1); 2. selection of L-proteins (in this case, monobodies) with high affinity for the D-target, 3.) chemical synthesis of the D-monobodies, which, by well established principles of chirality, will have equal affinities to the natural L-target that the corresponding L-monobodies have to the D-target.

The potential payoff of this labor-intensive approach is that D-monobodies would be predicted to have heightened plasma stability and resistance to degradation by proteases, as compared to L-monobodies, potentially furnishing protein therapeutics with superior pharmacokinetic properties.

Carrying out a prodigious amount of peptide synthesis, the authors show quite convincingly that the approach can work. They produce two D-monobodies (D-DAM21 and D-DAM27) and demonstrate their binding to the BCR::ABL1 SH2 domain. They also solve the crystal structures of the corresponding L-monobodies bound to the D-SH2. Somewhat frustratingly, the crystal structures show two different monobody/SH2 interfaces, yet the authors make a convincing case that they are able to determine which interface (interface 2) is the one relevant to the complex in solution.

On the whole, the biochemical experiments in the paper are clear and convincing. However, the manuscript contains several limitations:

We thank the referee for acknowledging the technical challenges of our work and the large amount of work that we invested. We are also happy that our biochemical experiments were described as clear and convincing. In the revised manuscript we provide additional data, which we hope will overcome the spotted limitations of our study together with the explanations and clarifications below.

1) Lack of monobody cell permeability: It is puzzling that the authors chose an intracellular target (SH2 domain of BCR::ABL1) as their target for their D-monobody approach, when there is no reason to expect that the monobodies would be cell permeable. Nothing can be done about this at this point, but the authors should discuss their target choice a bit more. Why did they choose the BCR::ABL1 SH2 when an extracellular protein domain would be the more intuitive choice for a D-monobody target?

The reviewer correctly identified some important issues that we are happy to comment about:

Lack of cell permeability: Our lab and others are assessing different methods to deliver monobodies to the cytoplasm of cells. One manuscript from our lab that is currently under review shows L-monobody delivery to cancer cells using the bacterial T3 secretion system. While highly efficient, this delivery method cannot be applied to D-monobody delivery as the monobody needs to be genetically encoded. We have been working over the past 3 years on

a second approach that would enable delivery of D-monobodies using polycationic resurfacing. However, we are still finishing work on fully characterizing delivery efficiency and improving endosomal escape. Hence it is not ready to be applied for D-monobody delivery yet. Furthermore, this would require adaptation in D-monobody synthesis, as a whole surface of the monobody needs to be re-engineered, which is a labour-intensive and time-consuming process and hence beyond the scope of the current study.

Extracellular protein domain would be the more intuitive choice: We agree that choosing an extracellular target would have been more straightforward. But as discussed in our manuscript, D-binders to extracellular proteins were previously developed (e. g. Mandal, Uppalapati et al., PNAS 2012, 109 (37), 14779-84). As a competing group was working on D-monobodies targeting extracellular MCP-1 (back-to-back submission with our manuscript), we decided to focus on an intracellular target. We believe that D-monobodies will only realize their full potential when used intracellularly, as besides the known benefits (low immunogenicity, high plasma stability), enhanced intracellular half-life and stability can be expected.

Choice of BCR::ABL1 as target: Our lab has been working on BCR::ABL1 for almost 20 years. It is also an established therapeutic target. We have long-term experience in developing and benchmarking high-affinity monobodies targeting the SH2 domain of BCR::ABL1 (Wojcik, Hantschel et al., NSMB 2010, 17 (4), 519-27; Grebien, Hantschel et al., Cell 2011 47(2), 306-19; Wojcik et al., JBC 2016, 291 (16), 8836-47) and other SH2 domains. Importantly, a variety of tools and characterization assays are available and the previously generated monobodies can be used as validated controls. Lastly, the chemical synthesis of D-Abl SH2 was previously established by us (Schmidt et al., RSC Chem. Biol. 2022, 3, 1008-12) and was therefore readily available for monobody selection.

Nevertheless, we understand that our reasoning may not have been obvious. **Hence, we now included a paragraph in the manuscript containing the arguments above (lines 98-101 and 116-124) and in the Discussion (lines 485-489).**

2) Lack of demonstrated selectivity by D-monobodies: The current manuscript contains no experiments that establish the selectivity for the BCR::ABL1 SH2 domain over other SH2 domains (or other non-SH2 protein domains, for that matter). The authors assume selectivity based on their selection, which utilized the SH2 from BCR::ABL1, but they should test this directly. Do D-DAM21 and D-DAM27 selectively bind BCR::ABL1 SH2 over other SH2 domains? If so, how selective are they? Of course, the authors need not measure binding constants for all other SH2 domains, but a few judiciously chosen SH2s (maybe the three most homologous?) would do the trick.

We thank the reviewer for bringing up this important point. We addressed selectivity using three approaches including the experiment suggested by the reviewer (point 2 below):

1. We first analyzed structural determinants for monobody selectivity. The Abl1 (and Abl2) SH2 domain has a particularly short CD loop which is a prerequisite for binding of DAM21 and DAM27. In contrast, all SH2s of the Src- and Tec-family (9 and 5 members, respectively), the closest paralogues of the Abl family, have CD loops that are 4-6 amino acids longer and would clash with the β C/ β D strands of DAM21 and DAM27 (**new Fig. 9A-C and SI Tab. S8**). Also, several other major SH2 families including the STATs, Jak kinases and SHP1/2 (C-SH2) have long CD loops. In addition, the ionic interaction of Glu51 and Arg189 important for monobody binding can only be formed with Abl kinase SH2 domains, as many other SH2 domains, including the Src kinase SH2s, do not have Arg (or Lys) in this position (**new SI Tab. S8**). Overall, these observations indicate a high selectivity of DAM21

and DAM27 for the Abl1 SH2 domain. **A figure containing structural representation and a sequence alignment were added to the manuscript and supplementary information (Fig. 9A-C and SI Tab. S8) and text describing this analysis (lines 378-391).**

2. To assess selectivity experimentally, we measured binding to the SH2 domains of the Lck and Btk tyrosine kinases, respectively, which are representative members of the Src and Tec kinases, the two most homologous tyrosine kinase families to the Abl family. No binding of D-DAM27 and D-DAM 21 was detected by isothermal titration calorimetry to both SH2 domains. **Figure and text were added (Fig. 9D-G; manuscript lines 391-395).**
3. To assess selectivity in an unbiased way using proteomics, we performed pulldown experiments comparing D- and L-DAM21/27 from K562 cells and subsequent analysis by mass spectrometry. Among SH2 domain-containing proteins, only BCR::ABL1 and a few of its known SH2-containing interactors (likely piggybacking on BCR::ABL1) were enriched with D-DAM21/27. This observation is in line with previous data with other ABL-targeting monoclonal antibodies (Wojcik et al., NSMB 2010, 17 (4), 519-57) and underlines the high selectivity of monoclonal antibodies in general. **These results were added to Fig. 9H-I, SI Tab. S9 and the manuscript lines 397-420.**

3) Uninformative experiments in permeabilized cells: As noted before, the authors' D-monoclonal antibodies are not cell permeable, so the authors must resort to using permeabilized cells to test whether the D-monoclonal antibodies can engage the target SH2 in a complex mixture of proteins (Figure 9C). Unfortunately, this experiment does not establish target engagement, as the fluorescence readout of the assay does not distinguish between on-target and off-target binding. In fact, off-target binding in this experiment would give higher signal, so a "big bar" does not provide evidence of binding to BCR::ABL1 SH2. Largely because this experiment does not distinguish between on-target and off-target binding, the data raise more questions than they answer: Why does the previously established Abl binder HA4 not give rise to a signal increase? Why do the L-DAMs (which shouldn't bind BCR::ABL1 SH2) give rise to signal increases? Why are the signal increases of the D-DAMs so much larger than those from the previously established BCR::ABL1 binders? (This reviewer's guess is that the huge increases for the D-DAMs is due to off-target binding.) In short, this experiment is uninformative. The figure should be removed, and the authors should figure out a better way to probe target engagement in a complex proteome. Maybe a better experiment would be to incubate immobilized D-monoclonal antibodies with a crude lysate and quantify the proteins that bind by mass spec. In addition to establishing that the D-monoclonal antibodies can bind their target in a complex proteome, this would help address the selectivity question raised above.

The main argument of the reviewer is the unknown selectivity which hampers the interpretation of the experiment. Given the additional data and analysis establishing high selectivity of D-DAM21/27 (see point 2 above and new **Fig. 9 + SI Tab. S8-S9**), we are therefore convinced that the signal observed with D-DAM21/27 results from on-target binding. In our opinion, the most relevant controls are the L-counterparts for which we observe a significantly lower signal. We have no good explanation why the HA4 monoclonal antibody does not seem to bind well in this assay. But it is important to consider that it has a different interaction mode and affinity/selectivity than D-DAM21/27. As the experiment contains relevant controls and we now provide evidence that D-DAM21/27 are selective, we do not see a strong argument to remove this experiment entirely, but we decided to **move it to the SI (SI Fig. S23)**.

As suggested by the reviewer to probe target engagement in a complex proteome, we performed quantitative proteomics experiments, in which we compared pulldowns of the D- and L-versions of DAM21/27 in triplicates. Both ABL1 and BCR peptides could be detected and quantified and were enriched with the D-versions of DAM21 and DAM27, respectively. Importantly, also several known and validated BCR::ABL1 interactors (likely piggybacking with

BCR::ABL1) were enriched with D-DAM27 and D-DAM21. These results strongly suggest binding of BCR::ABL1 complexes in the complex proteome of K562 cells. **The data is now included in Fig. 9H-I, SI Tab. S9 and the text was amended accordingly (lines 397-420).**

Referee #2

We all know about the possibility in life to create a mirror-image of proteins and ligands. While D-enzymes would recognise and modify D-target partners, the recognition of the L-target would become problematic. In this work Nina Schmidt et al. demonstrated the possibility to identify affinity reagents in the L-form that associate to a synthesised D-target protein with high affinity (11-70 nM). Thereafter as expected, they show that the D-affinity reagent also recognises the native L-target protein. This is important as the D-affinity reagent is expected to be stable in plasma and non-immunogenic, so as to become an improved therapeutic agent.

The use of a synthetic library of monodies (stable, small, monomeric domains lacking disulfide bonds, ability to refold from linear polypeptide) seems to be a clever choice, although DARPins or affibodies could have been valid alternatives.

The crystal structure of two L-monobodies in complex to the D-Abl-SH2 target has been solved and shows that the binding is overlapping with a D-pY peptide binding site.

The work has been explained in a logical and clear manner. Although there were apparently problems to synthesize the D-alternative of the affinity reagent, the authors found an elegant solution by generating a split variant.

In view of the variable and abundant amount of work shown, my critics to the work are minor:

We are thankful for the overall positive remarks of the reviewer about our manuscript and are happy to address the remaining points below.

1) The last part of the last sentence of the abstract "... enabling their use as future D-protein therapeutics to target a broad spectrum of protein-protein interactions" is not supported by the current work: First, the split monobody is probably too unstable to be used as a therapeutic. Secondly, more importantly, to have a biological effect they use a concentration of 5 microM added to permeabilised cells. This is still far from a practical solution. Nowhere was there an indication of how to get the synthesised D-protein inside cells. Perhaps it might be a possibility if the target recognition would occur on the cell membrane, rather than cytoplasmic.

We are sorry that this phrase led to misunderstandings about therapeutic use of monobodies at this stage. We have therefore toned down and re-wrote this sentence in the abstract (lines 42-44).

While we agree that low micromolar concentrations may be difficult to achieve therapeutically, it is not an uncommon concentration for binding/inhibition assays. In fact, for the binding assay in Fig. 9C (new Fig. S23) only 2 µM were used and due to the complexity of the experiments with its many controls, we did not attempt to titrate down monobody concentrations, but are convinced that binding and inhibition can be achieved with lower concentrations.

2) It was also confusing to note that the "split L-DAM21 showed inhibition as compared to its L-counterpart" (Line 352-353). Does this mean that the L-DAM21 binds to D-Abl-SH2 as well as to L-Abl-SH2??? or does it mean that neither of them bind sufficiently to have an impact on enzyme inhibition

(in contrast to D-DAM27). The binding study (Fig9C) suggests (to me) that indeed the L DAM21 doesn't bind significantly to its D target, in contrast to the binding of the split D-DAM21. Could it be that the split D-DAM21 is binding to the L target, but the D-DAM21 is too unstable to exert a kinase inhibition.

We are sorry for the confusion, but the reviewer seems to have incorrectly quoted the sentence by inadvertently swapping “L-” with “D-” when writing her/his review: Line 352-353 in our original manuscript read “split-monobody D-DAM21 [NB: not L-DAM21] showed inhibition as compared to its L-counterpart”, This is in line with the results shown in Fig. 9B (Fig. 8G in revised manuscript; compare bars 4 (dark purple) and 5 (light purple)), which showed a significant difference in kinase inhibition by D-DAM21 when compared to L-DAM21. This is also in line with Fig. 9C (Fig. S23 in revised manuscript), where D-DAM21 showed a much stronger binding signal in cells than L-DAM21 (compare bars 5 and 6). The confusion could also be caused by an incoherent colour coding of the kinase assays (Fig. 9A, B; revised manuscript Fig. 8F, G) and binding assay (Fig. 9C; revised manuscript Fig. S23). **We have adapted the colour coding in Fig. S23.**

Referee #3

This manuscript describes the identification and characterization of D-monobodies against the BCR-ABL1 SH2 domain. As a very small antibody-mimicking domain, monobodies are particularly well-suited for synthesis in mirror-image. The first step is synthesis of the mirror-image BCR-ABL1 SH2 domain (following their previous L-synthesis). Monobody libraries are then screened using a combination of initial phage display with polishing rounds via yeast display. 6 clones were identified and the two highest affinity clones were further characterized (by crystallography, ITC, competition FP). These two monobodies were synthesized in D (one by NCL & the other as a pair of complementing fragments) and further characterized, including proteolytic resistance and inhibition of kinase activity (in vitro and in permeabilized cells).

Overall, this is an interesting, thorough, and clearly presented paper that advances the case for D-monobodies as a promising emerging class of inhibitors. However, there are some additional controls that would strengthen the conclusions and some minor technical issues that should be addressed (listed below).

We are grateful that our manuscript is rated positively in general and for the useful comments.

1) it would be helpful to add a brief discussion about how a D-monobody will access an intracellular target, especially in the context of a mirror-image monobody (fully synthetic)

We agree that a section about cell permeability of monobodies should be included in the manuscript as our D-monobodies are targeting an intracellular protein. **We now included a paragraph in the manuscript containing this information (lines 98-101 and 116-124; lines 485-489).** Currently we are working on monobody delivery into the cytosol by generating polycationic monobody versions that are able to pass the cell membrane.

2) no details on the progression of library selection were provided - a brief summary in the supplement would be helpful (e.g., progress of rounds, sequencing details).

As this is a standard method and published before, we assumed that it is not necessary and only included references in the Methods section. **We now added a section about monobody selection in the supplement (SI, section 2).**

3) the case for specificity would be strengthened by adding controls to measure specificity towards the BCR-ABL1 SH2 domain (vs. other SH2 domains)

As suggested by the reviewer, the specificity was tested by measuring binding via isothermal titration calorimetry of the monobodies to the SH2 domains of Lck und Btk tyrosine kinases, respectively, which are representative members of the Src and Tec kinases, the two most homologous tyrosine kinase/SH2 families to the Abl family. No binding of D-DAM27 and D-DAM 21 was detected by isothermal titration calorimetry to both SH2 domains. **Figure and text were added (Fig. 9D-G; lines 391-395).**

Furthermore, the monobody selectivity was analysed based on the obtained crystal structures. The Abl1 (and Abl2) SH2 domain has a particularly short CD (β 3- β 4) loop which is a prerequisite for binding of DAM21 and DAM27. In contrast, all nine SH2 domains of the Src kinase family, the closest paralogues of the Abl family, have CD loops that are 4-6 amino acids longer and would clash with the β D/ β E sheet of DAM21 and DAM27 (**new Fig. 9A-C and SI Tab. S8**). Also, several other major SH2 families including the STATs, Jak kinases and SHP1/2 (C-SH2) have long CD loops. In addition, the ionic interaction of Glu51 and Arg189 important for monobody binding can only be formed with Abl kinase SH2 domains, as many other SH2 domains, including the Src kinase SH2s, do not have Arg (or Lys) in this position (**SI Tab. S8**). Overall, these observations indicate a high selectivity of DAM21 and DAM27 for the Abl1 SH2 domain. **A figure containing structural representation and a sequence alignment were added to the manuscript and supplementary information (Fig. 9A-C and SI Tab. S8) and text describing this analysis (lines 378-391).**

Lastly, to assess selectivity in an unbiased way using proteomics, we performed pulldown experiments comparing D- and L-DAM21/27 from K562 cells and subsequent analysis by mass spectrometry. Among SH2 domain-containing proteins, only BCR::ABL1 and a few of its known SH2-containing interactors (likely piggybacking on BCR::ABL1) were enriched with D-DAM21/27. This observation is in line with previous data with other ABL-targeting monobodies (Wojcik et al., NSMB 2010, 17 (4), 519-57) and underlines the high selectivity of monobodies in general. **These results were added to Fig. 9H-I, SI Tab. S9 and the manuscript lines 397-420.**

4) L-DAM21 & L-DAM27 bind surprisingly well (~50% of signal observed with D-DAM21/27) to BCR-ABL1 in permeabilized cells (despite showing minimal binding by BLI). It is unclear why this non-specific binding is so high - would be helpful if the authors could explain (or at least speculate).

We are happy to speculate why the binding of synthetic L-monobodies was higher than expected, however, we exclude the possibility that it might come from off-target binding of the synthetic monobodies as the pulldown experiment shows no detectable off-targets (see above). We assume that it might come from some unspecific interaction resulting in a suboptimal signal-to-noise ratio, because the monobody concentration applied to the cells and detection by Streptavidin-AlexaFluor488 was not titrated/optimized. Overall, it might be too high which leads to increased signals due to unspecific staining. Since the control monobodies have different sequences, affinities and were produced differently (bacterial expression vs. chemical synthesis), their staining efficiency and purity could be different and hence a much lower signal is detected even compared to the synthetic L-counterparts which were supposed to not bind. Therefore, the most relevant controls for D-DAM21/27 binding are the L-counterparts for which we observe a significantly lower signal. It is not uncommon in FACS-like staining experiments, even when using commercially available antibodies for staining, that the staining efficiency across binders, binder concentration used and FACS buffer composition

leads to detection of different signal intensities. Therefore, an optimization of the monobody concentrations and FACS buffer composition could improve the window of specific to unspecific staining and signal-to-noise ratio. Although we now provide evidence that D-DAM21/27 selectively bind BCR::ABL1 in cells, we decided to **move this experiment to the SI (SI Fig. S23)**, as also reviewer 1 commented about the experiment.

5) The D83A/E85A mutation control would have more value if the folding of this protein was confirmed (e.g., by CD).

The CD spectrum of the D83A/E85A mutant (compared to the wildtype protein) with analysis of the secondary structure content is now included in the supplement (see SI, Fig. S8 and Tab. S4). The two CD spectra are almost identical indicating that the D83A/E85A mutant protein is as well-folded as the wildtype protein.

6) line 736: I think “rippled” was meant instead of “scribbled”

We thank the reviewer for spotting this typo, which we **corrected (line 858)**.

7) FP curves (for Fig. 2F) should be shown in SI

We included a FP curve for Abl SH2 binding to the pYEEI peptide in the supplement (see SI, Fig. S1). Based on this binding curve we used a concentration of 10 μ M in the endpoint assay with monobodies where ~80% of the maximum binding signal of Abl SH2 to the peptide was observed, which ensures the sensitive detection of possible competitive inhibitors.

8) line 152: “elution volumes of all monobodies corresponded to their calculated monomeric molecular weights”, but DAM28 elutes at a higher molecular weight

It is correct that DAM28 elutes at a higher molecular weight. We do not have an explanation for that and it is one reason why we did not follow up on this monobody. **Hence, we rephrased the sentence to include this observation (line 163-164).**

9) In Fig. 6B, what is the source of the discrepancy between recombinant & synthetic DAM27-XTEN by SDS-PAGE?

While the amino acid sequences of the core monobody sequence are identical between recombinant and synthetic constructs, the recombinantly expressed DAM27-XTEN construct carries four additional amino acids at the N-terminal region (because of unavoidable additional bases/amino acids from cloning). For small proteins, such as monobodies, these seemingly minor differences in sequence length can result in a different migration distance. Furthermore, the synthetic constructs carry Biotin on the N-terminus which could influence their migration pattern in SDS-PAGE. **We make the readers aware of this by adding an explanatory statement in the manuscript (lines 306-310).**

10) In Fig. 7, what is the source of the significant discrepancies in stoichiometry? Impurities in the synthetic constructs?

We consistently observed in our ITC measurements that the stoichiometry was always smaller for synthetic constructs, which is also in line with our previous paper (Schmidt et al., RSC Chem. Biol. 2022, 3, 1008-12). We assume this comes from residual impurities, which may affect concentration measurement. As a result, the real concentration of the synthetic protein solution is lower which has an impact on stoichiometry. Based on the CD spectrum we can exclude that a significant fraction of the protein is unfolded, which would result in a discrepant stoichiometry in ITC.

11) In Fig. 8B, there are some irregularities on the gel: synth-L-DAM27-XTEN shows higher signal at 24 h than at 4 h & the D-DAM27-XTEN signal appears to be increased at 4 & 24 h (above input). Something similar is shown in Fig. 8C (for the D-protein).

The higher signal at 24 h for synthetic L-DAM27-XTEN in Fig. 8B comes from a smear in the gel. **The digest was repeated and no smear is visible this time, as expected. The figure panel was replaced with the new one (Fig. 7B in revised manuscript).** The increased signal for synthetic D-DAM27-XTEN in Fig. 8B and C (**new Fig. 7B-C**) might come from extended incubation of the samples at 37 °C. For both experiments this could have led to evaporation of water from the sample and increasing the concentration of the monobody leading to a stronger band in the gel and Western blot.

12) In Fig. 9B, the lack of inhibition seen with D-DAM21 is puzzling (inhibition compared to the slight stimulation seen with the L-DAM21 negative control is not convincing).

This observation was also puzzling to us and we discussed this result in great depths before submission of this manuscript. Firstly, it is important to consider, as shown in new Fig. 6E, that the split-DAM21 monobody is less stable than DAM27. Hence, it might not be too surprising that it has a weaker inhibitory effect. Secondly, there might be some confounding variables including the possibility that the control monobodies might get phosphorylated to a different extent by the kinase during the assay because they contain several tyrosine residues. Hence, we believe that the most relevant comparison is between (split)-L- and D-DAM21. As we reproducibly saw a lower activity when split-D-DAM21 was added to the kinase assay compared to split-L-DAM21, we are convinced that this can only stem from the binding of split-D-DAM21 to the Abl SH2 domain.

Point-by-point response to the reviewers' comments (NCOMMS-24-22129A)

Our comments in red, **additional data/changes to manuscript/figures in bold red**

Reviewer #1 (Remarks to the Author):

The authors were responsive to my comments on the previous manuscript, and the revised manuscript is significantly improved. In particular, the new experiments presented in Fig. 9D-G provide evidence for selectivity for the Abl SH2 domain over other SH2 domains, and I was pleased that the authors carried out the proteomic experiments that I suggested upon reviewing the original submission.

We are happy that the reviewer thinks the manuscript is significantly improved and that the additional data is providing evidence for the selectivity of our D-monobody binders.

Unfortunately, the proteomics results are disappointing and raise both technical questions and questions of interpretation that should be addressed in another round of revision.

Technical questions/issues:

- 1.) The authors state that the MS experiments were performed in triplicate (“Pulldown and mass spectrometry analysis were performed three times for each protein”), yet all the data are presented as single values without associated error bars. Are the presented data averaged between three experiments or taken from a single representative experiment? If averaged, the authors should include error bars and describe how the error values were calculated. If representative, the authors should state why they did not average and present the data from the other two experiments in Supplementary Information.

We apologize for not being clearer on experimental procedures and data analysis. We performed pulldowns in three independent replicates for each of the four monobodies (L-/D-DAM21/27) and all three replicates were used for data evaluation. Peak areas of unique peptides identified by mass spectrometry were calculated of every replicate, averaged, and plotted as \log_2 of the area ratio between the respective D- and L-monobody pairs. Furthermore, the *PEAKS12* (Bioinformatics Solutions Inc, Canada) analysis software performed a statistical analysis of the replicates and calculated a p-value that indicates if the two compared data sets are identical, which means that a lower p-value (and therefore higher $-10\log_{10}(p)$ -value) corresponds to the data sets being probably NOT identical. Therefore, identified proteins that appear further towards the upper right section in the volcano plots are enriched with the D-monobody.

The data and statistical analysis in the first revised submission was performed with *PEAKS12* (Bioinformatics Solutions Inc, Canada) which displays the $-10\log_{10}(p)$ -values as “significance”. Unfortunately, this software calculates significances for all samples

of the experiment (i.e. for D-/L-DAM21 and D-/L-DAM27 together) and displays only one value for all, which does not adequately reflect the intended comparisons between D-/L-DAM21 OR D-/L-DAM27 separately. Additionally, this reviewer's critic of the MS data made us analyze the dataset in more detail and we noticed that the software reported missing values for single replicates of several identified proteins including ABL1. Usually, the *PEAKS12* software quantifies the peak areas of identified proteins and performs the statistical analysis of the comparisons afterwards, where it uses peak area values of the 1-3 most quantifiable peptides. However, in this dataset it seemed to not have worked for several single replicates and the software reported missing peak area values, which we only realized when deep diving into the data. The missing peak area values seemed to have an impact on the p-value and significance calculations. Hence, after discussion with our mass spectrometry expert, we decided to change the data analysis and statistical analysis procedure of the raw data. By using the software packages *MaxQuant 2.5.1.0* (<https://www.maxquant.org/>) and *Autonomics* (R package version 1.13.21), it is possible to make pairwise comparisons of D-/L-DAM21 OR D-/L-DAM27 separately and calculate p-values for each identified protein for the pairwise comparisons (D- vs. L-DAM21 and D- vs. L-DAM27) resulting in more conventional-looking volcano plots, which in our opinion resemble the data comparisons more accurately.

As customary in reporting proteomics data as volcano plots, only the statistical analysis ($-10\log_{10}(p)$ -value, y-axis) and $\log_2(\text{ratio})$ (x-axis) are shown, which already take variability of the replicates into account. Hence, it is not necessary to report error bars.

In summary, we changed the analysis software to depict the complex data sets more appropriately and included updated volcano plots in the main manuscript (Fig. 9H-I), updated tables with corresponding values in the Supplementary Information (Tab. S9-S10) and changed the text of the manuscript discussing our data in a better way (lines 397-427). For more details on interpretation see point "Question of interpretation" below. We also revised the figure legend (lines 954-967) and methods section (lines 765-767) to more clearly explain the underlying analysis. Additionally, we deposited the mass spectrometry proteomics data to the ProteomeXchange Consortium via the PRIDE partner repository with the dataset identifier PXD056009. The dataset is not publicly accessible yet and can be seen by the reviewer through login by using the following account details:

Username: reviewer_pxd056009@ebi.ac.uk

Password: SZy8fdQpveMv

- 2.) How did the authors define and calculate “Significance” (y-axis in Fig. 9H-I)? Why is the significance identical for a protein between separate experiments? In other words, why are there single significance values for each protein (Table S9), as opposed to separate values for the DAM-21 and DAM27 experiments?

The “Significance” value was calculated by the *PEAKS12* software that we used for the original data analysis in the first revised manuscript: It represents the $-10\log_{10}$ of a p-value, which represents the likelihood that the observed change between conditions is caused by random chance. The software PEAKS provides the significance testing via ANOVA using a two-tail T-test that assumes log normal distribution but does not assume equal variance. A Significance score of 20 is equivalent to a p-value of 0.01. Since Significance is calculated with ANOVA across all groups, which means all measured samples together, only one Significance value is displayed for each protein to indicate how significant the change is in the full experiment. For graphical convenience, the representation was split in separate volcano plots for DAM21 and DAM27.

As mentioned above, we changed the data analysis and statistical analysis to *MaxQuant 2.5.1.0* and *Autonomics* (R package version 1.13.21) to analyze the comparisons between D-/L-DAM21 and D-/L-DAM27 separately and to generate volcano plots from the analysis with GraphPad Prism (Fig. 9H-I) which depict the data sets more appropriately. Additionally, we used the imputation function within *Autonomics* to not generate missing values for replicates, which makes the calculations more reliable.

- 3.) The authors refer to the proteomic experiments as occurring “in cells” when the experiments were performed in cell lysates. This should be fixed prior to publication. (For example, “Binding of D-monobodies to Bcr-Abl in CML cells: In order to determine if D-monobodies can bind to full-length BCR::ABL1 in the context of the complex proteome of human cells, we performed...”)

We thank the reviewer for pointing this out and rephrased all affected sentences and titles to “in cell lysates” (lines 40-41; 397; 400; 434-435; 955).

Question of interpretation:

The text in lines in lines 397-420 does not really engage the proteomic data shown in Fig. 9H-I and Table S9, which reveal a disappointing lack of D/L selectivity. The data are actually quite troubling in regard to the manuscript’s central hypothesis that the D-DAMs selectively bind to Abl in the context of a complex proteome. The data in Table S9 show that D-DAM27 and L-DAM27 pull down able Abl in essentially equal amounts (D/L ratio = 1.09).

Since our analysis procedure changed, we changed the text (lines 397-427) and volcano plots (Fig. 9H-I) accordingly and hope that it engages the complex proteomic

data more appropriately. In summary, the data provide a significant enrichment of BCR (15.9-fold) and ABL1 (3.4-fold) peptides in the pulldown with D-DAM21 (Table S9). The pulldown with D-DAM27 shows significant enrichment of BCR (3.6-fold) as well, but only a mild enrichment of ABL1 (1.4-fold, but not significant; Table S10). Given the low abundance of BCR::ABL1 in cells (~10,000 protein copies in K562), as compared to the hundreds of highly abundant housekeeping proteins/core proteome, a single step pull-down is a challenging experiment and not expected to result in extremely high enrichment values without extensive optimization, which is cost-prohibitive for a proteomics experiment. Hence, we are satisfied with the obtained results, because a preferential binding of D- over L-DAM21/27 is demonstrated. Since BCR::ABL1 is a fusion oncoprotein, either identification/enrichment of BCR or ABL1 peptides is sufficient to claim binding of the D-monobodies to BCR::ABL1. While we do not have a sound explanation for the low enrichment of ABL1 peptides with D-DAM27, the data indicates selective and significant enrichment of several BCR::ABL1 interactors (1.9-5.3-fold, Table S10 and see below) with D-DAM27 (as well as for D-DAM21). Together with the lack of binding affinity of L-DAM27 and L-DAM21 for the L-Abl SH2 domain (Figure S22), we believe that these arguments support our conclusion that full-length BCR::ABL1 is bound preferentially by both D-monobodies. **As suggested below, we rephrased the text in the Results section.**

Likewise, a few of the known Abl interactor proteins have D/L ratios of approximately 1 and some even have ratios of less than 1, indicating greater pull down with an L-monobody than the corresponding D-monobody (e.g., ABI, AB2M1, CBLB, CRKL). The authors should completely re-write the description of the proteomic experiments in a manner that accurately describes the selectivity complications revealed by the data.

To exclude bias, we used the curated *BioGRID* biological database of protein-protein interactions (<https://thebiogrid.org/>) to identify validated BCR::ABL1 interactors after the re-analysis of our proteomics dataset with *MaxQuant* and *Autonomics* (see above). Several BCR::ABL1 interactors were found for both D-monobodies with significant enrichment, which underline the fact that ABL1 is pulled down selectively by the D-monobodies. Additionally, other proteins containing SH2 domains, which were not published as ABL1 interactors, were not enriched and hence support the selectivity of the D-monobodies towards ABL1. **We therefore rephrased the text to point out our new findings (lines 409-421).**

Reviewer #2 (Remarks to the Author):

The authors took the comments and critics of the reviewers into consideration and replied in the best possible way.

We are pleased to have answered the reviewer's points sufficiently.

Reviewer #3 (Remarks to the Author):

This revised manuscript has addressed the concerns raised in the original review (with extensive new supporting data, as well as additional caveats in the text). The addition of experimental specificity data is especially valuable. It remains a substantial body of work and, with the revisions, the conclusions are now robustly supported.

We are happy that we were able to address all concerns raised by the reviewer.